# Loss of HDAC11 ameliorates clinical symptoms in a multiple sclerosis mouse model

Lei Sun[1,2,*], Elphine Telles[1,*], Molly Karl[3,*], Fengdong Cheng[1], Noreen Luetteke[1], Eduardo M Sotomayor[1], Robert H Miller[3], Edward Seto[1,2]

Multiple sclerosis (MS) is a chronic, immune-mediated, demyelinating disease of the central nervous system (CNS). There is no known cure for MS, and currently available drugs for managing this disease are only effective early on and have many adverse side effects. Results from recent studies suggest that histone deacetylase (HDAC) inhibitors may be useful for the treatment of autoimmune and inflammatory diseases such as MS. However, the underlying mechanisms by which HDACs influence immune-mediated diseases such as MS are unclear. More importantly, the question of which specific HDAC(s) are suitable drug targets for the potential treatment of MS remains unanswered. Here, we investigate the functional role of HDAC11 in experimental autoimmune encephalomyelitis, a mouse model for MS. Our results indicate that the loss of HDAC11 in KO mice significantly reduces clinical severity and demyelination of the spinal cord in the post-acute phase of experimental autoimmune encephalomyelitis. The absence of HDAC11 leads to reduced immune cell infiltration into the CNS and decreased monocytes and myeloid DCs in the chronic progressive phase of the disease. Mechanistically, HDAC11 controls the expression of the pro-inflammatory chemokine C–C motif ligand 2 (CCL2) gene by enabling the binding of PU.1 transcription factor to the CCL2 promoter. Our results reveal a novel pathophysiological function for HDAC11 in CNS demyelinating diseases, and warrant further investigations into the potential use of HDAC11-specific inhibitors for the treatment of chronic progressive MS.

## Introduction

Multiple sclerosis (MS) is a chronic demyelinating disease that affects more than two million people worldwide (Zurawski & Stankiewicz, 2017). This disease is characterized by progressive inflammatory demyelination within the central nervous system (CNS), leading to motor deficits and cognitive and sensory impairment. Most MS patients initially experience a relapsing-remitting course of disease, characterized by immune attack and demyelination of axons, followed by total or incomplete remyelination (Fletcher et al, 2010). Over time, remyelination fails and the disease becomes chronic, characterized by slowly increasing functional deficits. There is no known cure for MS and most current therapies mediate immune suppression or immune modulation, which is predominantly effective in relapsing-remitting MS, but there are currently no effective treatments for the chronic disease.

Experimental autoimmune encephalomyelitis (EAE) is one of the most commonly used animal models for the study of MS. EAE induces a T cell–mediated autoimmune reaction to myelin antigens which is characterized by the infiltration of the CNS with macrophages and lymphocytes (Tompkins et al, 2002; Kawakami et al, 2004). C57BL/6 mice are common models for EAE induction using myelin oligodendrocyte glycoprotein (MOG) peptides because of their predictable responses and wide availability of transgenic and KO mice in this strain background. EAE in C57BL/6 mice is usually manifested as a chronic disease. In general, immunization with MOG peptide 35–55 ($MOG_{35-55}$) results in a monophasic EAE with initial symptoms after 9–14 d, and maximal symptom severity at about 3–5 d after disease onset. The disease course is generally chronic, although slow and partial recovery may occur over the next 10–20 d (Bittner et al, 2014).

The anti-inflammatory property of histone deacetylase inhibitors (HDACi) has been exploited in both preclinical and clinical studies to treat inflammatory diseases, including colitis induced by dextran sulphate or trinitrobenzene sulphonic acid, Crohn's disease, and T cell lymphoma (Camelo et al, 2005; Glauben et al, 2006; Mann et al 2007a, Mann et al 2007b). HDACi have also been used to protect neurons from oxidative stress, modulate the growth/survival of neurons and oligodendrocytes (Faraco et al, 2011), and treat neurological disorders such as epilepsy and mood swings (Tunnicliff, 1999). The neuroprotective and immunosuppressive

[1]George Washington University Cancer Center, George Washington University School of Medicine and Health Sciences, Washington, DC, USA  [2]Department of Biochemistry and Molecular Medicine, George Washington University School of Medicine and Health Sciences, Washington, DC, USA  [3]Department of Anatomy and Cell Biology, George Washington University School of Medicine and Health Sciences, Washington, DC, USA

Correspondence: seto@gwu.edu
*Lei Sun, Elphine Telles, and Molly Karl contributed equally to this work.

effects of HDACi suggest that HDACi may potentially be useful for treatment of neuroinflammatory diseases including MS. For example, the two broad-spectrum HDACi trichostatin A (TSA) and valproic acid, as well as Vorinostat (which preferentially inhibits class I and HDAC6, although it is not highly selective), have been shown to ameliorate EAE (Camelo et al, 2005; Zhang et al, 2012; Ge et al, 2013; Pazhoohan et al, 2014; Lillico et al, 2018). However, the nonspecific nature of these inhibitors possibly contributes to the heterogeneous and suboptimal therapeutic outcomes (Dietz & Casaccia, 2010). Therefore, a comprehensive analysis of each histone deacetylase (HDAC) to determine its individual functions in inflammation and MS is essential to evaluate specific HDAC targets for optimal use of HDACi as potential MS treatments.

In humans and mice, there are 18 HDACs that are divided into four classes based on their homology with yeast HDACs. HDAC11 belongs to the class IV family, and shares a highly conserved deacetylase domain with other family members (Gao et al, 2002; Glozak et al, 2005; Yang & Seto, 2008; Seto & Yoshida, 2014). Human HDAC11 mRNA is highly expressed in the brain, heart, kidney, and skeletal muscle (Gao et al, 2002). Early studies indicate that HDAC11 regulates the expression of interleukin 10 and immune tolerance (Villagra et al, 2009), whereas a number of recent studies confirm that HDAC11 possesses immune regulatory functions (Huang et al, 2017; Sahakian et al, 2017; Woods et al, 2017; Yanginlar & Logie, 2017).

Although little is known about the neurological functions of HDAC11, results from an early study showed that of the 11 classical HDACs, HDAC11 is expressed most highly throughout the rat brain (Broide et al, 2007). The authors argue that the presence of HDAC11 in oligodendrocytes suggests a role for HDAC11 in myelination that may be better understood in the context of demyelinating conditions such as MS. In other studies, HDAC11 expression was found to correlate with oligodendrocyte-specific gene expression, including myelin basic protein (MBP) and proteolipid protein, suggesting that HDAC11 may modulate oligodendrocyte development and function (Liu et al, 2009; Jagielska et al, 2017). Finally, higher HDAC11 transcript levels were reported in female patients with MS, implicating a clinical significance of HDAC11 in this neurological disease (Pedre et al, 2011).

Here, we examine the role of HDAC11 in CNS demyelinating conditions by comparing the severity and pathology of EAE in KO versus WT mice. We found that a lack of HDAC11 does not affect the initiation and development of the autoimmune response in EAE. Rather, HDAC11 appears to regulate the later chronic progressive phase of EAE. HDAC11 KO mice display amelioration of paralytic disease symptoms in the post-acute phase of EAE, accompanied by reduced inflammation and demyelination of the spinal cord and decreased infiltration of monocytes and DCs. Experiments exploring the molecular mechanism reveal that HDAC11 stimulates chemokine CCL2 expression by affecting the binding of PU.1 transcription factor to the CCL2 promoter. Hence, loss of HDAC11 reduces secretion of CCL2 and migration of inflammatory cells, which ultimately leads to alleviation of chronic EAE symptoms in KO mice. The results of this study indicate that HDAC11 contributes to neuroinflammation in EAE and suggest that targeting HDAC11 holds promise for the treatment of chronic progressive MS.

# Results

## HDAC11 KO mice have reduced disease severity and develop fewer clinical symptoms in chronic post-acute phase of EAE

HDAC11 KO mice were generated by the deletion of exon 3 flanked with loxP sites, resulting in a frame-shift mutation that leads to the loss of HDAC11 protein expression. Genotyping of mouse-tail DNAs performed by using PCR with the indicated primers (Fig S1A and B). RT–PCR reveals a shorter product in multiple organs of HDAC11 KO mice compared with WT animals, consistent with exon 3 deletion (Fig S2A). Results from in vitro translation of HDAC11 full-length cDNA compared with that of missing exon 3 eliminates the possibility of alternative HDAC11 proteins (Fig S2B). Loss of HDAC11 protein expression was further confirmed by immunoblotting protein extracts from WT and HDAC11 KO mouse brains (Fig S2C). To rule out the possibility that the loss of HDAC11 could lead to a compensatory increase in expression of other classical HDACs, HDAC mRNA levels were evaluated in WT and HDAC11 KO brains. No statistically significant differences in the mRNA levels of class I and class II HDACs were observed between HDAC11 KO and WT mice (Fig S2D).

The development and lifespan of HDAC11 KO mice are normal, and compared with WT mice, there was no abnormal behavior or appearance in HDAC11 KO mice. Moreover, body weights and brain weights between WT and HDAC11 KO female mice at 20 wk had negligible differences (Fig S3A and B). Histopathology of hematoxylin and eosin (H&E) staining showed that there were no obvious differences in brain anatomy between the two groups (Fig S3C).

To test our hypothesis that HDAC11 is involved in the development or progression of CNS demyelinating diseases, we immunized WT and HDAC11 KO C57BL/6 mice with $MOG_{35-55}$ to induce EAE. As shown in Fig 1A, both WT and HDAC11 KO mice developed clinical EAE, and the disease onset (around 11 d post-immunization) and peak acute clinical disease symptoms (around 14 d post-immunization) were nearly identical in both genotypes. However, in the ensuing chronic progressive disease phase, beyond around 20 d post-immunization, the clinical severity was significantly lower in HDAC11 KO mice. In the control WT group, one mouse died during the course of a typical experiment, and many mice had to be euthanized by 40 d post-immunization because of their moribund condition. By contrast, none of the HDAC11 KO mice died or had to be euthanized for the entire duration of the experiments. The mean overall clinical score for HDAC11 KO mice was 2.4, which was significantly lower than 3.3 for WT mice (Table 1). These findings suggest that the lack of HDAC11 does not significantly affect the initial induction of EAE, but that HDAC11 modulates the later chronic phase of the disease.

## HDAC11 KO mice have less demyelination and less immune cell infiltration into the spinal cord

Because EAE is an inflammatory demyelinating disease of the CNS, we examined the extent of demyelination in mouse spinal cords at various stages. Transverse sections of spinal cords isolated from mice at 0, 14, 27, and 39 d post-immunization were stained with Luxol Fast Blue (LFB). As expected, no significant difference in white matter distribution or myelin staining intensity was observed in WT

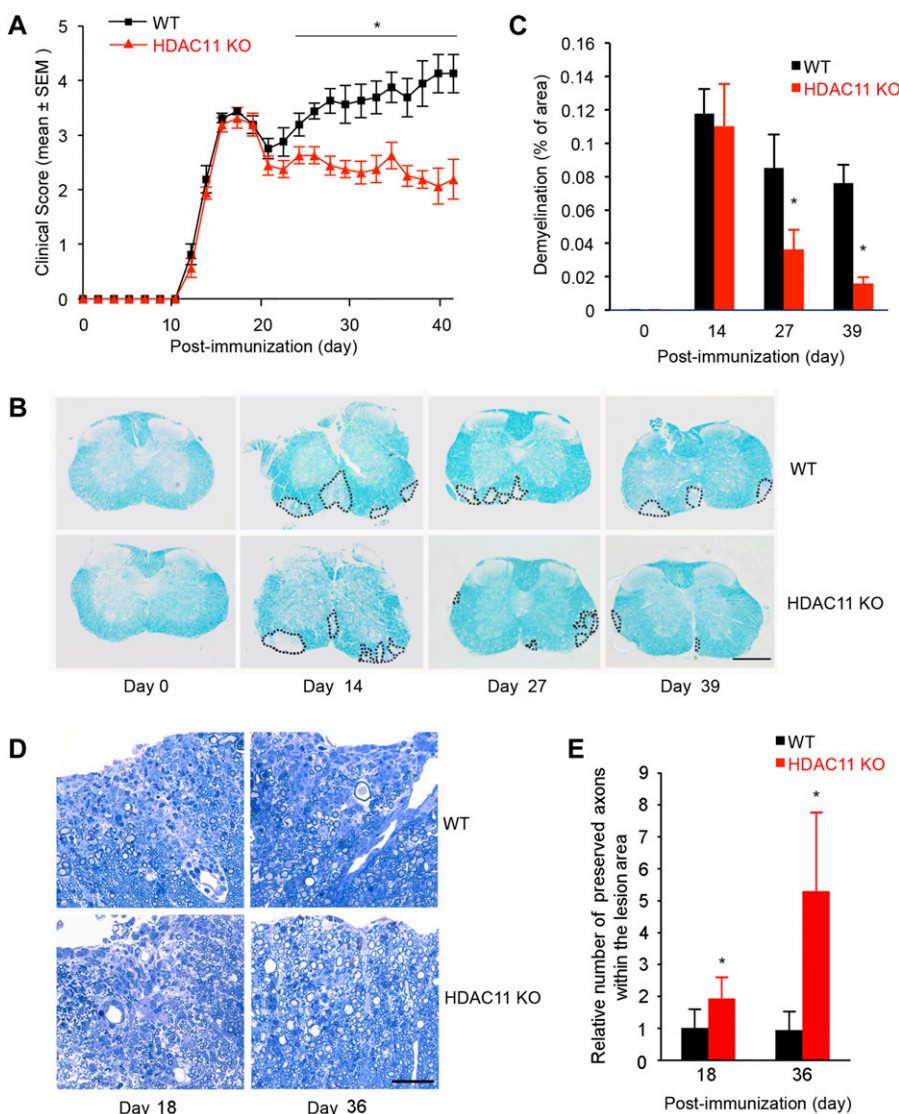

Figure 1. **Loss of HDAC11 promotes functional recovery in the chronic phase of MOG$_{35}$–55 induced EAE.**
**(A)** Clinical scores of EAE in C57BL/6 WT (n = 10) and HDAC11 KO (n = 10) mice immunized with MOG$_{35-55}$. Results are presented as the mean clinical score ± SEM in each genotype during the course of each experiment. **(B)** Lesion load determined by area of spinal cord demyelination was assessed by LFB staining. Representative histological sections from 0, 14, 27, and 39 d post-immunization are shown, and the area of demyelination of the spinal cord is delineated by dots. Scale bar, 400 $\mu$m. **(C)** Quantitative comparison of relative lesion load (demyelination) between WT and HDAC11 KO mice. **(D)** Representative images of Toluidine blue stained spinal cord lesions, 18 and 36 d post-immunization. Scale bar, 100 $\mu$m. **(E)** Quantitative comparison of relative number of preserved axons within the lesion area between WT and HDAC11 KO mice. Data shown are mean ± SD from more than three samples per time point, and $P$-values were calculated by $t$ test, *$P < 0.05$.

versus HDAC11 KO mice before immunization (day 0) (Fig 1B and C). Consistent with the observation that the acute clinical disease symptoms are nearly identical between WT and HDAC11 KO mice, an

equivalent level of demyelination was apparent at 14 d post-immunization in the spinal cords of both WT and HDAC11 KO mice. However, at 27 and 39 d post-immunization, HDAC11 KO mice clearly displayed approximately 40–80% reduction in the areas of spinal cord demyelination than WT mice, consistent with their lower clinical severity scores in the later chronic progressive disease phase. In addition, Toluidine Blue staining of spinal cord sections obtained from EAE mice 18 and 36 d post-immunization further confirmed that HDAC11 KO mice had more preserved axons compared with WT groups within the lesion areas (Fig 1D and E). Taken together, these data suggest that HDAC11 deficiency permits functional CNS recovery via remyelination in the chronic phase of EAE.

During induction of EAE, mononuclear inflammatory cells infiltrate the CNS, contributing to ascending paralysis of the mice. H&E stained sections of mouse lumbar spinal cords reveal a significant increase in inflammatory cells in both the WT and HDAC11 KO mice post-immunization (comparing day 0 to day 14, Fig 2A). The

**Table 1. Clinical EAE signs in HDAC11 WT and HDAC11 KO mice after immunization with MOG$_{35-55}$.**

|  | HDAC11 WT | HDAC11 KO |
|---|---|---|
| Number of animals (n) | 10 | 10 |
| Percent incidence | 100% | 100% |
| Day of onset | 13.2 ± 0.4 | 13.3 ± 0.5 |
| Clinical score | 3.3 ± 0.7 | 2.4 ± 0.5 |
| Peak disease severity | 4.1 ± 0.9 | 2.1 ± 1.0 |

Disease onset is defined by the second consecutive day in which the animals showed a clinical score of at least 0.5. Results for day of onset, clinical score, and peak of disease severity are shown as mean ± SEM ($P < 0.05$; Mann–Whitney $U$ test).

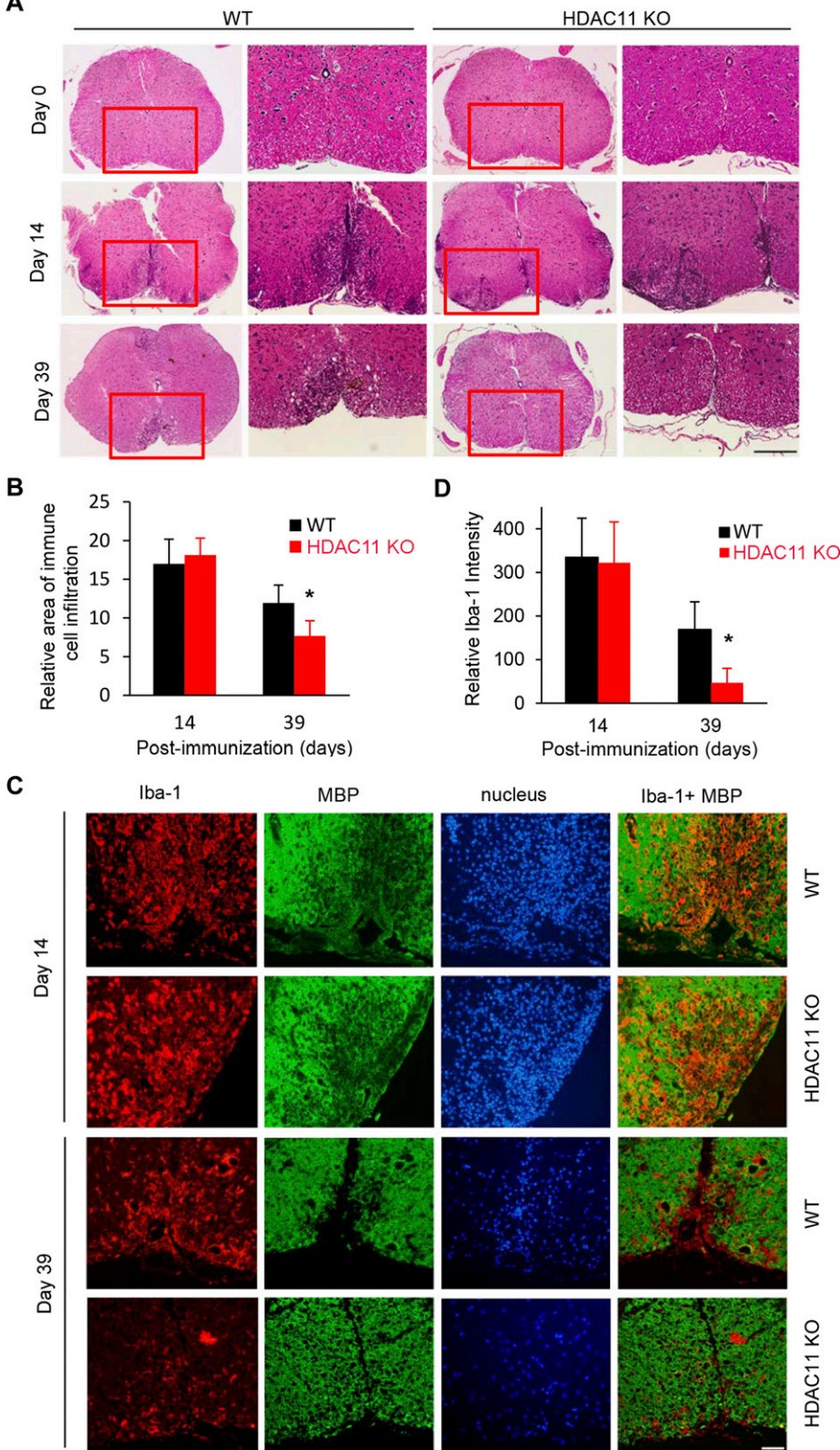

**Figure 2. Reduced immune cell infiltration in spinal cords of HDAC11 KO compared with WT mice.**
**(A)** Representative images of H&E-stained lumbar spinal cords, isolated from WT and HDAC11 KO mice on days 0, 14, and 39 post-immunization. Scale bar, 200 μm. **(B)** Quantitative comparison of relative area of immune cell infiltration between WT (n = 3 per group) and HDAC11 KO (n = 3 per group) mice. **(C)** Representative fluorescence photomicrographs of anti-Iba1 staining to assess activated microglia/macrophages, and anti-MBP staining to assess myelin sheaths in spinal cords of EAE animals. Iba1: red; MBP: green. Scale bar, 50 μm. **(D)** Quantitative comparison of relative Iba-1 staining intensity between WT (n = 3 per group) and HDAC11 KO (n = 3 per group) mice. Data shown are mean ± SD, and P-values were calculated by t test, *P < 0.05.

presence or absence of HDAC11 has no discernible influence on the amount of inflammatory infiltrates during the early acute disease phase. By contrast, immune cell infiltrates in HDAC11 KO mice were visibly reduced compared with WT mice 39 d post-immunization, consistent with reduced disease severity in HDAC11 KO mice in the chronic progressive disease phase of EAE (Fig 2A and B).

To further assess demyelination in WT and HDAC11 KO EAE mice, spinal cord sections from these animals were stained with antisera against MBP, a major component of CNS myelin. Although comparable MBP immunofluorescence was detected in WT versus HDAC11 KO mice at day 14 post-immunization, a much lower level of myelin perturbation was detected in HDAC11 KO compared with WT mice at day 39 post-immunization (Fig 2C). Thus, both LFB and MBP staining also suggest that remyelination of the spinal cord during the chronic progressive phase of EAE is significantly higher in the absence of HDAC11.

Activation of microglia commonly occurs in response to a wide variety of pathological stimuli in the CNS. Ionized calcium-binding adaptor molecule-1 (Iba1), which is specifically expressed in CNS microglia and macrophages, has been shown to be up-regulated upon activation of microglia. Immunohistochemistry for Iba1 was performed to assess the infiltration of peripheral macrophages and activation of microglia in the spinal cords of WT and HDAC11 KO mice with EAE. At the peak of acute disease on day 14 post-immunization, a large number of Iba1$^+$ cells are present in both the WT and HDAC11 KO mouse spinal cords, particularly in demyelinated (MBP negative) areas (Fig 2C). However, compared with the WT mice, spinal cords from HDAC11 KO mice contain significantly fewer Iba1$^+$ cells on day 39 post-immunization (Fig 2C and D). These results confirm a consistent correlation between improved clinical severity scores, better spinal cord remyelination, and lower infiltration and activation of macrophages/microglia during the chronic progressive disease phase of EAE in HDAC11 KO compared with WT mice.

Because MS is often accompanied by neurodegenerative disorders, we compared the number of motor neurons in the grey matter of EAE mice 36 d post-immunization between WT and HDAC11 KO groups. Our results indicate no obvious difference in motor neurons between the two groups (Fig S4A). Likewise, the amount of non-phosphorylated neurofilament H assessed with anti-SMI-32 staining in the grey matter reveals no significant difference between WT and HDAC11 KO EAE mice (Fig S4B).

To further support the hypothesis that the effects of HDAC11 on demyelination and disease severity are exerted via the immune system, we used the cuprizone diet model of MS (Matsushima & Morell, 2001). In this model, demyelination induced by feeding mice a diet containing cuprizone is a result of oligodendrocyte apoptosis, and does not involve the adaptive immune response. As shown in Fig S5, although there appears to be a small reduction of myelin in WT versus HDAC11 KO mouse brains, the rate and extent of demyelination and remyelination were comparable in the cuprizone model of these animals.

Another method to study nonimmune-mediated demyelination is by focal injection of lysophosphatidylcholine (LPC) into the mouse spinal cord to produce a lesion that could be spontaneously repaired over time. To determine whether HDAC11 plays a role in this demyelination and remyelination model, L-$\alpha$-lysophophatidylcholine (lysolecithin) was injected into the dorsal column of HDAC11 KO and WT mouse spinal cords. 5 and 10 d after injection, LPC-induced demyelination and subsequent remyelination in WT and HDAC11 KO animals were compared. At 15 d post LPC injection, only a modest reduction in the size of spinal cord lesions in the HDAC11 KO mice was observed compared with the WT mice (Fig S6A and B). Based on these results, the role of HDAC11 in MS most likely involves

immune responses, although the effect of HDAC11 in oligodendrocyte function cannot be ruled out.

## Loss of HDAC11 leads to reduced peripheral monocytes and myeloid DCs (mDCs)

Development of EAE results from the activation and expansion of myelin specific T cells in the peripheral lymphoid organs followed by translocation to the CNS (Brown et al, 1982; Izikson et al, 2002). To test if reduced immune infiltration in the CNS reflected a diminished peripheral activation, we examined the peripheral autoimmune response to MOG. Splenocytes from WT and HDAC11 KO mice were stimulated with MOG and were analyzed for the immune cell types by flow cytometry. Compared with WT mice, HDAC11 KO mice had a significantly lower percentage of CD11b$^+$ CD4$^-$ monocytes (36.0 WT versus 19.2 KO) and CD11b$^+$ CD11c$^+$ mDCs (8.7 WT versus 4.4 KO). Also, although there was a slight increase in percentage of CD4$^+$ T cells (56.9 WT versus 66.8 KO), a considerable reduction in the percentage of CD8$^+$ T cells (30.8 WT versus 20.6 KO) was observed for HDAC11 KO mice (Fig 3A–C and Table 2).

Mononuclear cells isolated from spinal cords also showed a reduction in the percentage of mDCs induced by MOG peptide immunization in HDAC11 KO versus WT mice (2.46 ± 0.7 versus 31.4 ± 2.5; Fig 3D). We then compared the mRNA levels of CD11b, CD4, and CD8 in these immune cells from EAE mice. CD11b was chosen as a marker for inflammation as it is expressed in infiltrating macrophages, DCs, NK cells, and CD8$^+$ T cells (Bullard et al, 2005). Consistent with lower encephalitogenic response in the periphery, ~40–60% lower levels of CD11b and CD8, but not CD4, transcripts were observed in the spinal cords of HDAC11 KO mice (Fig 3E–G). These results indicate that the loss of HDAC11 leads to impaired recruitment of monocytes, DCs, and CD8$^+$ T cells to the CNS without compromising effector CD4$^+$ T cell infiltration.

We next determined if the decrease in immune infiltration into the spinal cords of HDAC11 KO mice was associated with a diminished peripheral response to MOG$_{35–55}$ peptide. To evaluate the peripheral autoimmune response, mice were immunized with MOG/CFA emulsion at the base of the tail. 12 d later, lymph nodes were isolated from HDAC11 KO and WT mice and re-challenged with increasing concentrations of MOG peptide in vitro. HDAC11 KO cultures showed a significant reduction in dose response of stimulation by MOG compared with WT cells (Fig 4A). CD4$^+$ and CD8$^+$ T cells isolated from mice of both genotypes exhibited equivalent activation in vitro, as indicated by their ability to secrete similar amounts of IFNγ in response to MOG peptide stimulation (Fig S7).

## Loss of HDAC11 reduces chemokine (C–C motif) ligand 2 (CCL2) production

Cytokine profiling of splenocyte cultures prepared from EAE mice showed strikingly lower levels of chemokine ligand 2 (CCL2), also known as monocyte chemoattractant protein-1 (MCP-1), in HDAC11 KO mice compared with WT mice (Fig 4B and C). Although HDAC11 was reported to negatively regulate IL-10 expression in the mouse macrophage cell line RAW264.7 and in human APCs (Villagra et al, 2009), no significant difference in IL-10 secretion was observed between the two genotypes of mouse splenocytes. Marked decreases

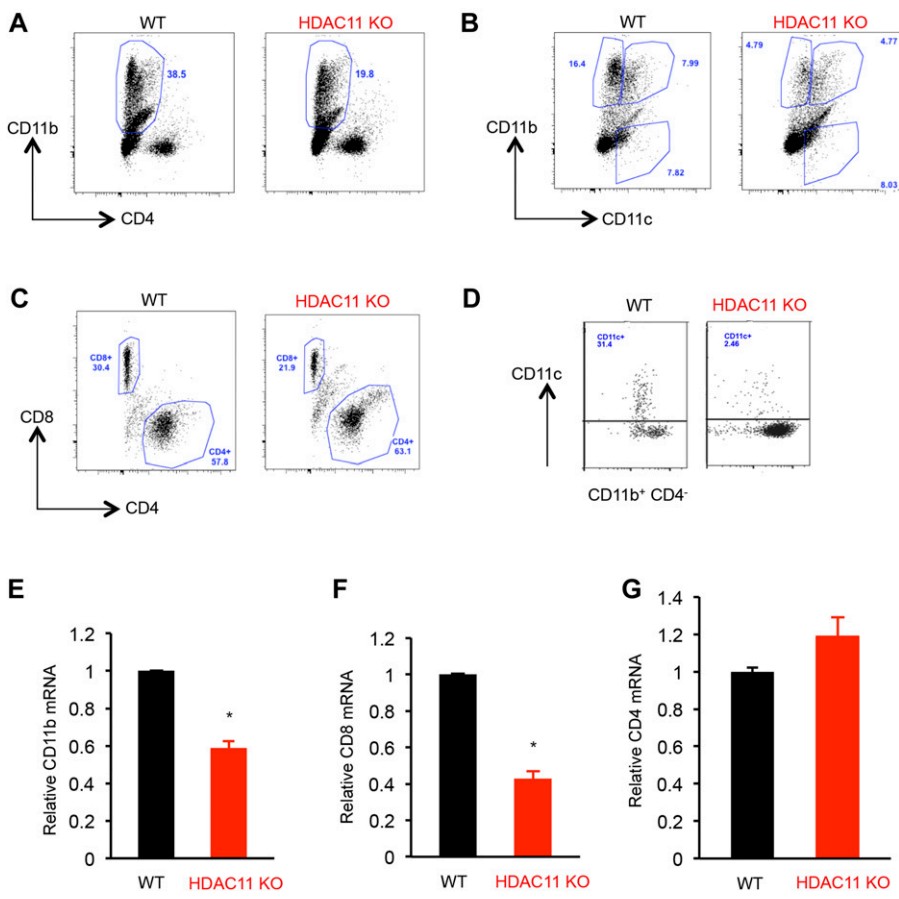

**Figure 3. Fewer monocytes and mDCs in HDAC11 KO mice.**

**(A–C)** Flow cytometric analyses of splenocytes isolated from WT and HDAC11 KO mice 12 d after MOG$_{35-55}$ peptide immunization. Spleen cells were isolated and cultured with 20 μg of MOG$_{35-55}$ peptide for 72 h and were restimulated with phorbol 12-myristate 13-acetate (PMA) plus ionomycin, with the addition of Brefeldin A for the last 4 h. The cells were stained with antibodies to CD11b-Cy5, CD4-Pacific Blue, CD11c-FITC, or CD8-APC. Analyses were gated on live cells based on live/dead yellow staining. **(D)** Flow cytometric analyses of mononuclear cells from spinal cords of WT and HDAC11 KO EAE mice, 21 d after MOG$_{35-55}$ peptide immunization. **(E–G)** Quantitative polymerase chain reaction (qPCR) analyses of CD11b, CD8, and CD4 mRNA expression in inflammatory cells from spinal cords isolated from WT and HDAC11 KO EAE mice 40 d after MOG$_{35-55}$ peptide immunization. Data shown are mean ± SD from three independent animals per genotype, and P-values were calculated by t test, *P < 0.05.

in levels of CCL2 transcripts by approximately 40–90% were also observed in the spinal cords of HDAC11 KO mice compared with WT mice during the chronic phase of EAE (Fig 4D). The reduction in CCL2 transcripts was more pronounced in spinal cords isolated 40 d post-immunization, which corresponds to reduced disease severity in the later phase of EAE.

CCL2, a cytokine of the CC chemokine family, has been demonstrated to recruit monocytes to active sites of inflammation. The reduction in immune cell infiltration in the spinal cords of HDAC11 KO mice could be attributed to decreased CCL2 production, which is consistent with previous findings from CCL2 KO mice and selective deletion of CCL2 in astrocytes (Huang et al, 2001; Kim et al, 2014). Taken together, our results suggest that a key function of HDAC11 is to regulate CCL2 expression.

**Table 2. Splenic cell infiltrates in HDAC11 WT and HDAC11-deficient mice after immunization with MOG$_{35-55}$.**

|  | HDAC11 WT | HDAC11 KO |
|---|---|---|
| CD11b+CD4− (%) | 36 ± 3.1 | 19.2 ± 0.84[a] |
| CD11b+CD11c+ (%) | 8.7 ± 0.6 | 4.4 ± 0.3[a] |
| CD8 (%) | 30.8 ± 0.2 | 20.6 ± 1.7[a] |
| CD4 (%) | 56.9 ± 0.9 | 66.8 ± 3.1 |

Results from three independent experiments are shown as mean ± SEM.
[a]P < 0.05; t test.

## HDAC11 deficiency reduces CCL2 secretion and impairs peritoneal exudate macrophages' (PEMs) migration

Because CCL2 has chemotactic activity for monocytes and is implicated in the pathogenesis of several diseases characterized by monocytic infiltrates, we compared the ability of supernatants of splenocyte cultures prepared from WT and HDAC11 KO mice with EAE to induce macrophage migration in a modified Boyden chamber assay. In vitro experiments using PEMs isolated from HDAC11 KO or WT mice treated with lipopolysaccharides (LPS) showed that the loss of HDAC11 leads to decreased CCL2 transcripts and secretion of CCL2 (Fig 5A and B). Associated with lower CCL2 levels in HDAC11 KO mice, a slower migration of PEMs was observed in supernatants from HDAC11 KO splenocyte cultured media compared with HDAC11 WT (Fig 5C and D).

## HDAC11 activates CCL2 transcription by affecting the binding of PU.1 transcription factor to the CCL2 promoter

The CCL2 promoter is well characterized and contains multiple binding sites for NF-κB and PU.1 transcription factors (Aung et al, 2006). PU.1 binds to the PU-box, a purine-rich DNA sequence (5′-GAGGAA-3′), and is a transcriptional activator that may be specifically involved in the differentiation or activation of macrophages (Zhang et al, 1994). It has been reported that PU.1 is a major regulator of CCL2 gene expression because PU.1 KO mice are deficient in CCL2 secretion, and PU.1 KO cells

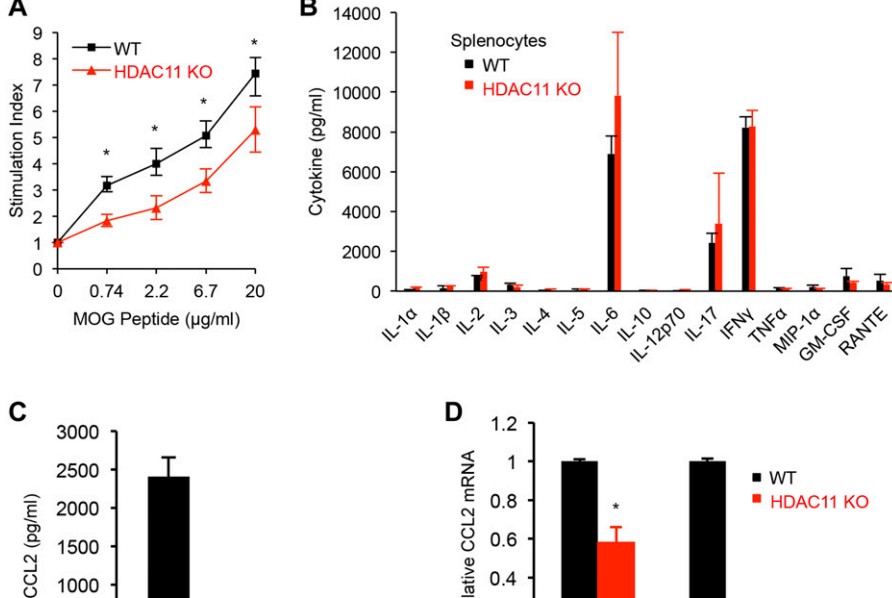

**Figure 4. Proliferation and cytokine production in MOG$_{35-55}$ immunized WT and HDAC11 KO mice.**
**(A)** Mice were immunized at the base of the tail, and lymph nodes were harvested 11 d later. Cells were cultured with increasing concentrations of MOG$_{35-55}$ peptide for 72 h and pulsed with thymidine during the last 14 h of culture. Stimulation index is calculated by dividing mean counts per minute ± SD of stimulated cells cultured with peptide to mean counts per minute ± SD of unstimulated cells cultured without peptide (n = 6). **(B, C)** Mice were immunized at two sites on the back, and spleens were harvested and cultured with 20 μg MOG$_{35-55}$ for 48 h. Supernatants were analyzed for cytokine production. Data shown are mean ± SD from three independent animals per genotype, *$P$ < 0.05. **(D)** Mice were immunized with MOG$_{35-55}$, mRNA was extracted from the lumbar region of the spinal cords at the indicated time post immunization (P.I.), and CCL2 mRNA expression was quantified by qPCR. Data shown are mean ± SD from three independent animals per genotype, of three independent experiments, and $P$-values were calculated by $t$ test, *$P$ < 0.05.

show reduced CCL2 expression (Karpurapu et al, 2011). PU.1 typically partners with other proteins to either activate or repress gene transcription (Suzuki et al, 2003). In co-immunoprecipitation assays with Flag epitope–tagged HDAC enzymes, we found that HDAC11 binds PU.1 (Fig 6A). In addition, consistent with a previous report, PU.1 interacts with HDAC1 and HDAC2 (Suzuki et al, 2003). However, the extent of PU.1-HDAC1/HDAC2 interaction is much lower relative to the association of PU.1 with HDAC11.

The PU.1 protein contains three functional domains: transactivation, PEST, and ETS DNA-binding domain (Fig 6B). To identify the domain in PU.1 that interacts with HDAC11, deletion mutants of PU.1 were tested for binding to HDAC11 via immunoprecipitation. As shown in Fig 6A and B, although full-length PU.1 and constructs that contain the ETS domain bind to HDAC11, deletion of the ETS domain eliminated the interaction of PU.1 with HDAC11. Therefore, the ETS domain of PU.1 is necessary and sufficient for interacting with HDAC11. By contrast, full-length HDAC11 binds PU.1, but small deletions of either N- or C- terminal sequences abolished the interaction, suggesting that PU.1 binds to multiple domains in HDAC11 (Fig 6C).

Because HDAC11 binds to the ETS DNA-binding domain of PU.1, we hypothesized that HDAC11 affects the interaction of PU.1 with DNA at the promoter of the CCL2 gene, and consequently alters CCL2 transcription to regulate inflammation. We first examined whether HDAC11 or PU.1 could directly regulate CCL2 expression. As shown in Fig 7A, qRT–PCR analysis showed that transfection of either HDAC11 or PU.1 cDNAs, increases the CCL2 mRNA level. A HDAC11 H143A mutant, which is catalytically dead in both defatty-acylase and deacetylase activities (Byun et al, 2017; Kutil et al, 2018), affects HDAC11 interaction with PU.1 and CCL2 expression (Fig 7B and C). We

also did not find any change in PU.1 expression upon over-expression of HDAC11 (Fig S8).

For further analysis, a CCL2 promoter containing nucleotides, −2,505 to +82 relative to the transcription start site (P-2505), and deletion constructs containing promoter fragments, −1,005 to +82 (P-1005) and −322 to +82 (P-322), were inserted into the luciferase reporter vector pGL3-Basic (Fig 7D). Both HDAC11 and PU.1 expression vectors activated transcription from P-2505, and the truncated CCL2 promoters containing PU.1 binding sites (Fig 7E). In the presence of the PU.1 binding sites, the extent of CCL2 promoter activation by HDAC11 and PU.1, as measured by luciferase activities, was nearly identical.

Using a −140 to +55 bp region of the CCL2 promoter that contains the PU.1 binding sites for chromatin immunoprecipitation (ChIP) assays, PEMs from HDAC11 KO mice showed reduced PU.1 binding to the CCL2 promoter compared with WT PEMs (Fig 7F). It has been reported that the deletion of CCL2 in astrocytes had less severe EAE and less macrophage and T cell inflammation in the nervous system (Kim et al, 2014). We, therefore, treated human glioblastoma astrocytoma U251 cells with MOG$_{35-55}$. Overexpression of PU.1 increased CCL2 expression. However, knockdown of HDAC11, decreased CCL2 expression more significantly in the presence of PU.1 overexpression (Fig 7G and H). These results strongly support the conclusion that HDAC11 regulates CCL2 expression mediated through PU.1.

## Discussion

Most people with MS are initially diagnosed with the relapsing-remitting form of the disease. In most patients, the disease advances into a chronic progressive phase and continues to worsen. While

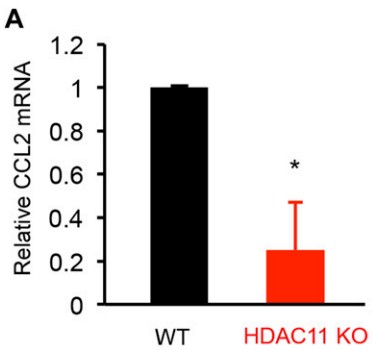

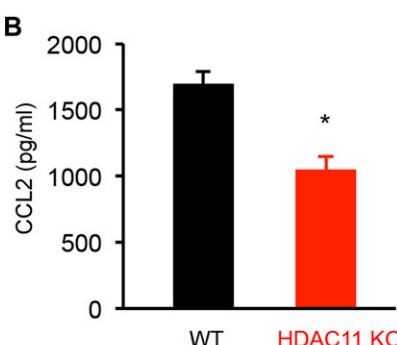

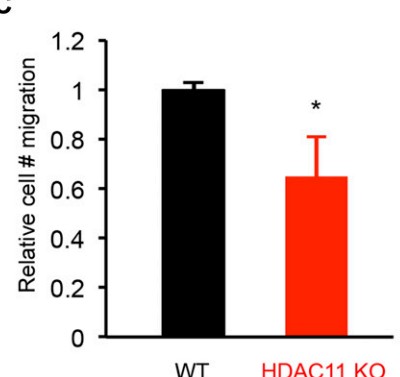

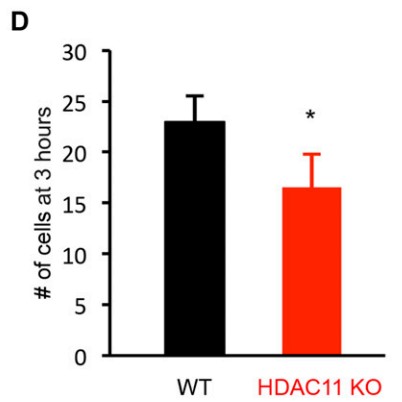

**Figure 5. CCL2 expression and migration of macrophages from WT and HDAC11 KO mice.**
**(A)** Equal numbers of thioglycollate-elicited murine PEMs were isolated from WT (n = 3) and HDAC11 KO (n = 3) mice and cultured in the presence of 1 μg/ml of LPS for 24 h. PEMs were harvested and mRNA was isolated for analysis by qPCR using the CCL2 primers listed in Table S1. **(B)** PEM culture supernatants were analyzed for CCL2 secretion by ELISA. **(C, D)** For cell migration assays, PEMs were isolated from WT mice and cultured in Transwell membrane inserts with medium conditioned for 48 h by either WT or HDAC11 KO splenocytes cultured with MOG peptide. The splenocytes were isolated from WT and HDAC11 KO mice with EAE. The number of PEMs migrating across the membranes at different times was counted as an average of three culture wells for each experimental group. Data shown are mean ± SD of three independent experiments, and P-values were calculated by t test, *$P < 0.05$.

treatment of relapsing-remitting MS has improved dramatically over the last decade, there are fewer treatment options available for chronic progressive MS, and current treatments are often limited to merely managing symptoms for these patients. In this study, using EAE animal models, we uncover a molecular mechanism for progressive MS and present evidence that inhibiting HDAC11 is a potential treatment strategy for chronic progressive MS.

HDACs were originally identified as enzymes that catalyze the removal of acetyl moieties from the ε-amino groups of conserved lysine residues in the amino terminal tail of histones. Later, it was discovered that HDACs have non-histone substrates and possess functions beyond chromatin structure and transcriptional regulation (Glozak et al, 2005). More recently, some HDACs have been found to possess other enzymatic activities in addition to deacetylation (Bheda et al, 2016). Unlike many HDACs, very little is known regarding the biological functions of the unique class IV enzyme, HDAC11. A phylogenetic tree analysis suggests that HDAC11 diverged from other class I and II HDACs early in evolution and may represent a key branch point that distinguishes class I and class II enzymes (Gregoretti et al, 2004). Notably, cell type and tissue distributions of HDAC11 are considerably different from those of most HDACs (Gao et al, 2002). Recently, it was reported that HDAC11 is a fatty acid deacylase rather than a deacetylase (Kutil et al, 2018; Moreno-Yruela et al, 2018), further arguing that HDAC11 might possess many unique and remarkable characteristics that are clearly not shared by other known HDACs.

A previous study reported an increase in HDAC11 transcripts in the normal-appearing white matter of the frontal lobes of MS

patients (Pedre et al, 2011). The differences were most prominent in a subset of female MS patients and were associated with high levels of developmentally regulated genes. The expression of several other histone-modifying genes, including HDAC1 and HDAC3 was unchanged. These results suggest that HDAC11 may potentially play a pathological role in MS patients. Based on this premise, a loss or inhibition of HDAC11 could have a protective effect in MS patients. In the current study, we investigated this hypothesis and found that HDAC11 KO mice have fewer clinical disease symptoms, less demyelination, and less CNS immune cell infiltration in the chronic progressive disease phase of EAE, a mouse model of MS.

EAE is a T cell–mediated, autoimmune disease of the CNS that is characterized by mononuclear cell infiltration and demyelination of the spinal cord leading to ascending paralysis. Loss of HDAC11 in KO mice reduced the overall population of CD11b+ cells, including CD11b+ CD4− monocytes and CD11b+ CD11c+ DCs during the later phase of EAE. Because CD11c+ DCs are sufficient for antigen presentation to T cells and disease development (Greter et al, 2005), no defect in initiation of an autoimmune response to MOG in HDAC11 KO mice was observed. In addition, the presence of high levels of IFNγ, IL-6, and IL-17 cytokines in the CNS and periphery indicate a robust autoimmune response similar to that of WT mice in the early phase of EAE.

HDAC11 deficiency also lowered CD8+ T cells in EAE mice. Although it is widely accepted that CD4+ T cells contribute to the pathogenesis of EAE, the role of CD8+ T cells is being increasingly recognized (Tompkins et al, 2002). Neuroantigen-specific CD8+

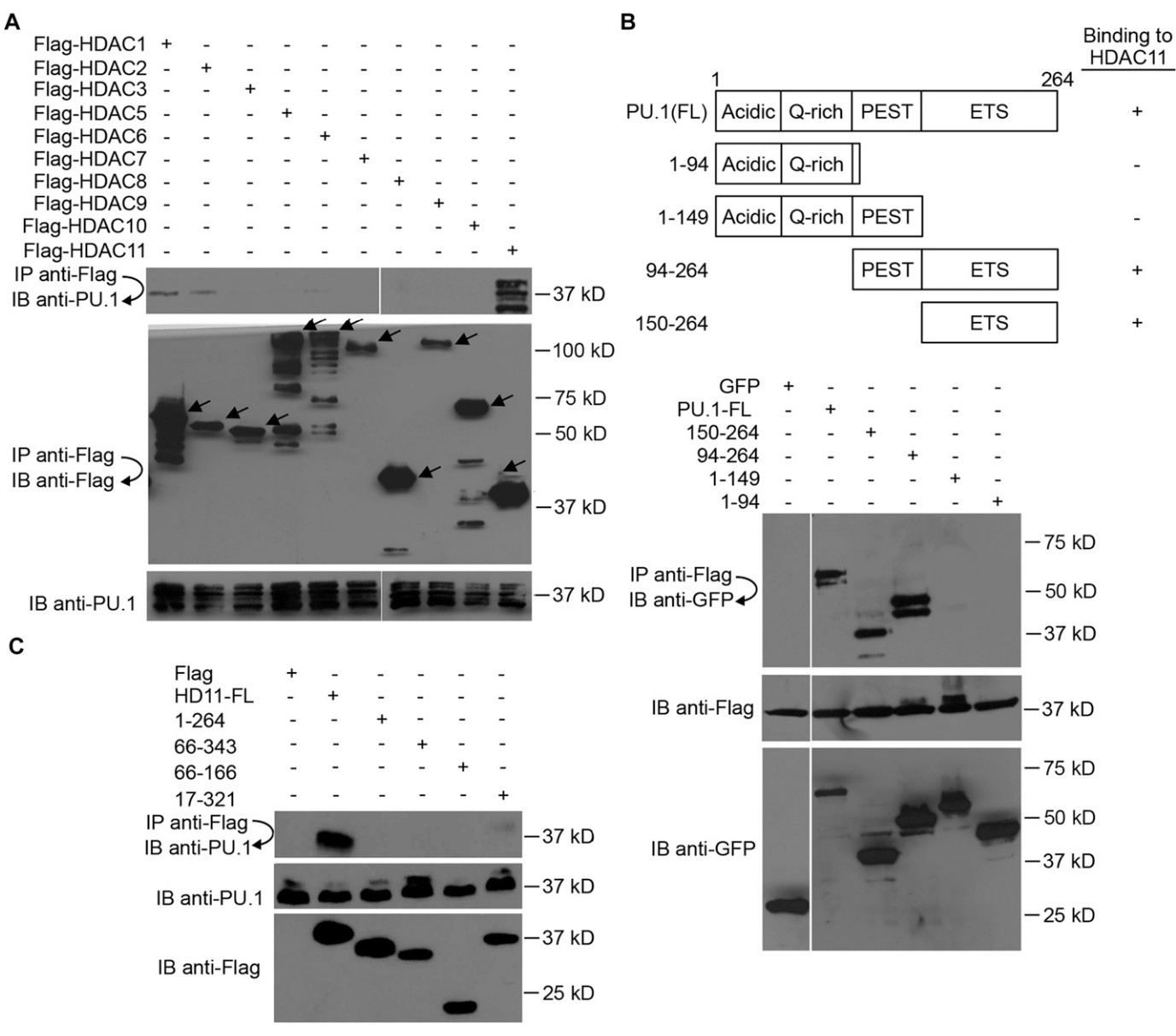

**Figure 6. HDAC11 interacts with PU.1.**
**(A)** 293T cells were transfected with PU.1 and Flag-tagged HDAC expression plasmids, followed by immunoprecipitation (IP) and immunoblotting (IB) as indicated. Arrows indicate the expected position of HDAC protein bands. **(B)** 293T cells were transfected with Flag-tagged HDAC11 and plasmids that express various truncated forms of PU.1 tagged with GFP. Flag antibody immunoprecipitation (IP) and GFP antibody IB were then performed to characterize the PU.1 domain required for interaction with HDAC11. **(C)** 293T cells were transfected with PU.1 and plasmids that express the indicated truncated forms of HDAC11 tagged with Flag. Flag antibody IP and PU.1 antibody IB were then performed to characterize the HDAC11 domain required for interaction with PU.1. Protein molecular weight size markers are shown on the right side of each panel. FL, full-length; acidic, acidic transactivation domain; Q-rich, glutamine-rich transactivation region; PEST, proline-, glutamine-, serine-, and threonine-rich domain; ETS, the Ezb transformation-specific sequence DNA binding domain.

T cells were abundant in the CNS of MS patients and a higher prevalence of CD8[+] T cell responses was noted in patients with relapsing-remitting MS (Babbe et al, 2000; Crawford et al, 2004). Also, CD8[+] T cells were more potent than CD4[+] T cells in inducing EAE in mice (Sun et al, 2001) and, like CD4[+] T cells, may also mediate epitope spreading (Bailey et al, 2007; Ji et al, 2013). Epitope spreading is the major determinant of relapsing EAE (Bailey et al, 2007) and is also observed in MS patients. Recent studies indicate that the primary T cell found in the nervous system in patients with

MS is the CD8[+] T cell, as patient-derived effector CD4[+] T cells are difficult to effectively discern from regulatory T cells (Tregs), which reduce the response of effector T cells and prevent autoimmune disease (Kaskow & Baecher-Allan, 2018).

In this study, we discovered that the loss of HDAC11 results in a drastic reduction in CCL2 levels, which was accompanied by decreased infiltration of monocytes and DCs into the spinal cords of EAE mice. CCL2 is among the most studied members of the chemokine family, and has been targeted as a potential intervention

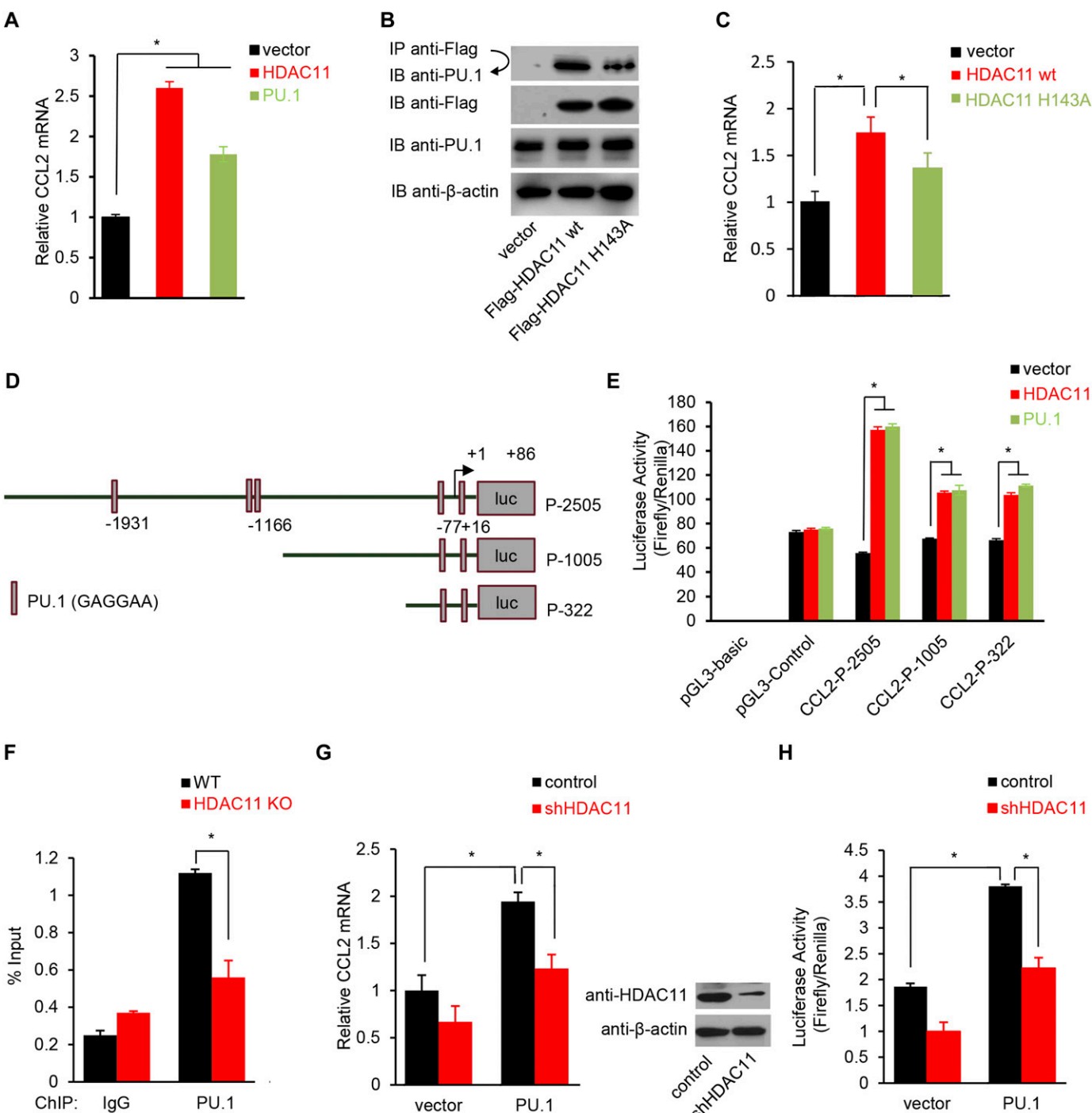

**Figure 7. HDAC11 regulates CCL2 expression by recruiting PU.1.**
**(A)** Real-time RT–PCR analysis of CCL2 mRNA expression in NIH 3T3 cells transfected with plasmids expressing PU.1, HDAC11, or an empty vector. GAPDH served as an internal control. **(B)** 293T cells were cotransfected with Flag-tagged vector, HDAC11 WT or mutant HDAC11 H143A, and PU.1 plasmids. Flag antibody immunoprecipitation (IP) and PU.1 antibody IB were performed to determine whether PU.1 interacts with mutant HDAC11. Flag-tagged HDAC11 H143A plasmid was generated from Flag-tagged HDAC11 using the Quikchange II XL site-directed mutagenesis kit (Agilent Technologies). **(C)** Real-time RT–PCR analysis of CCL2 mRNA expression in NIH 3T3 cells transfected with empty vector, or plasmids expression HDAC11 WT or HDAC11 H143A. GAPDH served as an internal control. **(D)** Schematic representation of luciferase (luc) reporter plasmids for various portions of the CCL2 promoter. **(E)** NIH 3T3 cells were transfected with indicated expression and luc reporter plasmids. Luciferase activity was measured in extracts 48 h after transfection. Data are expressed as mean ± SD of triplicates and are representative of at least two independent experiments. **(F)** ChIP analysis of PU.1 enrichment at the CCL2 promoter upon deletion of HDAC11. PEMs isolated from HDAC11 KO and WT mice were subjected to ChIP assays with the indicated antibodies. Data were normalized to input samples for the amount of chromatin. Normal IgG was used as negative control. **(G, H)** U251 cells with stable knockdown of HDAC11 expression (shHDAC11) or control were stimulated with $MOG_{35-55}$ peptide, then transfected with PU.1 plasmid (PU.1) or empty vector (vector). 48 h later, real-time RT–PCR analysis of CCL2 mRNA expression, and luciferase reporter assays of CCL2 promoter (CCL2-P-322) activity were performed. Immunoblots show stable expression of HDAC11 in U251 cells. All data represent mean ± SD from three experiments, and P-values were calculated by t test, *P < 0.05.

point for treatment of various diseases, including MS (Sørensen et al, 2004). CCL2 binds to the CCR2 receptor and stimulates the recruitment of monocytes to sites of inflammation. CCL2 mRNA is significantly up-regulated in EAE, and its expression is correlated with increased disease severity and relapse (Kennedy et al, 1998; Juedes et al, 2000). The importance of CCL2 in promoting relapse was further demon-strated by a decrease in disease severity in relapses but not in the acute phase when mice with EAE were administered anti-CCL2 at remission (Kennedy et al, 1998). Loss of CCL2 expression in the CNS resulted in lower accumulation of iNOS and TNF-expressing mac-rophages and mDCs. Expression of CCL2 coincided with onset of EAE after leukocyte infiltration into the CNS and correlated with relapsing disease, implying a role for CCL2 in amplifying the CNS inflammatory response (Glabinski et al, 1995; Kennedy et al, 1998).

Our studies demonstrate a significant reduction in CCL2 secretion during the chronic progressive phase in HDAC11 KO mice which correlated with lower disease severity scores, and less inflammation and demyelination in spinal cords. These findings indicate that HDAC11 may regulate CCL2 expression to promote progression of chronic EAE, and is consistent with a previous report that there is a limited role of CCL2/CCR2 in early active MS patients (Sørensen et al, 2004). It has been reported that PU.1 expression is blocked by HDACi (Laribee & Klemsz, 2001; Aung et al, 2006). However, we did not find any change in PU.1 expression upon overexpression of HDAC11, which indicates that PU.1 enrichment at the CCL2 promoter is not due to an increase in PU.1 expression. A HDAC11 enzyme-deficient mutant affects the interaction of HDAC11 with PU.1 and CCL2 expression, indicating that the enzymatic activity of HDAC11 is important for regulating CCL2 expression, and further suggesting that HDAC11 activates and reprograms CCL2 chemokine gene expression outside of the conventional role of HDACs in transcriptional repression.

Our results also demonstrate that HDAC11 promotes the ex-pression of CCL2 by directly binding to the ETS domain of the transcription factor PU.1 and affecting PU.1-DNA interaction at the CCL2 promoter. There are several, non–mutually exclusive, possible mechanisms that could explain the requirement of HDAC11 for the binding of PU.1 to the CCL2 promoter. First, recent reports indicate that HDAC11 is a potent fatty acid deacylase (Kutil et al, 2018; Moreno-Yruela et al, 2018). Conceivably, PU.1 undergoes fatty ac-ylation, and fatty acid deacylation of PU.1 by HDAC11 could indirectly lead to changes in PU.1 binding to DNA. Second, PU.1 binding could be directly regulated by posttranslational acetylation, and is po-tentially deacetylated by modest HDAC11 deacetylase activity. Al-though we have so far failed to detect changes in PU.1 acetylation levels in the presence and absence of HDAC11, these negative results could simply reflect the lack of a specific acetylated-PU.1 antibody. Finally, the acetylation/deacetylation of NF-κB could regulate its binding to the CCL2 promoter. Changes in NF-κB binding, adjacent to PU.1 binding sites, could in turn potentially regulate recruitment of PU.1 to the CCL2 promoter.

Neurodegeneration, either as an independent process or sec-ondary to inflammation, may also be a contributor to clinical outcomes in progressive MS. Based on our results, however, it is unclear if neurodegeneration might be a major cause of reduced clinical symptoms in the chronic phase of EAE in HDAC11 KO mice. What is clear is that our results are consistent with the beneficial effect of HDACi in mitigating clinical disease in EAE mice, and their

proposed neuroprotective, neurotrophic, and anti-inflammatory properties. Of note, the course of disease progression observed in HDAC11 KO mice with EAE matched that of TSA-treated EAE mice, wherein clinical amelioration was observed only during the chronic progressive phase (Camelo et al, 2005). However, nonselective HDACi, particularly TSA, have basal toxicity and prolonged treat-ment at high doses often induces neuronal death and antagonizes their neuroprotective effects. In fact, based on preclinical data, nonselective, broad-spectrum HDACi might exert detrimental ef-fects, contributing to MS pathogenesis. The results presented in the current study provide rationalization for the development of highly selective HDAC11 inhibitory drugs to treat chronic progressive MS.

This study confirms and extends previous reports of the im-portance of HDACs in CNS myelination/demyelination and MS. HDAC1 and SIRT1 have been reported to play a protective role in EAE mice (Rafalski et al, 2013; Tegla et al, 2014; Goschl et al, 2018). Furthermore, nuclear HDAC1 is exported to the cytoplasm in cuprizone-induced animal models of demyelination and in human brains with MS (Kim et al, 2010). Inactivation of SIRT1, a class III NAD-dependent deacetylase, increases the production of new oligo-dendrocyte progenitor cells and improves remyelination in EAE mouse models (Rafalski et al, 2013). However, the clinical outcome of SIRT1 KO is quite different from that of HDAC11 KO mice. Unlike HDAC11, which has no effect on the initial induction of EAE and selectively modulates the later chronic phase of the disease, SIRT1 inactivation delays the onset of paralysis in chronic EAE.

In summary, the observation that HDAC11 deficiency does not lead to any adverse phenotype in mice, combined with the con-vincing findings that the loss of HDAC11 ameliorates clinical symptoms in the chronic progressive disease phase of EAE, war-rants further investigations into the potential use of HDAC11-specific inhibitors for the treatment of chronic progressive MS.

# Materials and Methods

### Mice and EAE

*Hdac11* constitutive KO mice were generated at the Institut Clinique de la Souris (ICS) in collaboration with Merck & Co. and obtained via Taconic. These mice were backcrossed with the C57BL/6 back-ground for successive generations and then bred to homozygosity. WT C57BL/6 mice were used as controls for these experiments. To induce EAE, WT and HDAC11 KO female mice between 8 and 12 wk of age were immunized with $MOG_{35-55}$ peptide emulsified with CFA (Sigma-Aldrich)-containing *Mycobacterium tuberculosis* (200 ng/ml; Difco). For each animal, 100 $\mu$l of emulsion per site was depos-ited subcutaneously in the upper and lower backs. In addition, each animal was administered pertussis toxin (islet-activating protein, 500 ng, i.p.; List Biological Laboratories) at the time of immunization and again 1 d later. Mice were evaluated daily for changes in overt signs of illness and clinical signs of EAE using the following scoring system: 0, no physical signs; 0.5, distal tail limpness; 1, full-tail limpness; 2, mild or unilateral hind limb paresis; 3, full bilateral hind limb paralysis; 4, moribund; and 5, death attributable to EAE. Mice were euthanized if they scored four or above for 2 consecutive days. At least 10 mice per group were

used for each experiment, and all experiments were repeated three times. All experimental protocols were approved by The Institutional Animal Care and Use Committee.

## Plasmids and antibodies

Flag-tagged HDAC expression plasmids were previously described (Radhakrishnan et al, 2015). Similarly, PU.1 expression plasmids and PU.1 deletion constructs were previously described (Zhong et al, 2005). HDAC1 antibody was described (Tsai et al, 2000). Flag (clone M2) and HA (SAB1305536) antibodies were purchased from Sigma-Aldrich. IgG control (sc-2027) antibody was purchased from Santa Cruz Biotechnology. MBP (SMI 99) antibody was purchased from BioLegend. Anti-Iba1 (ab5076) was purchased from Abcam. To generate an anti-HDAC11 antibody, a peptide corresponding to amino acids 179–193 of HDAC11 (DLDAHQGNGHERDFM) coupled to keyhole limpet hemocyanin was injected subcutaneously into New Zealand white rabbits. The resulting antibody was immunoaffinity purified on a peptide column with Affi-gel agarose (Bio-Rad). Specificity of this anti-HDAC11 antibody was confirmed using brain tissue extracts from HDAC11 KO and WT mice.

## Histology and immunostaining

Mice were anesthetized with ketamine-xylazine solution and perfused intracardially with PBS through the left ventricle. Spinal cords were dissected and fixed in 4% formalin for 24 h, embedded in paraffin and sectioned at 8 $\mu$m. The sections were stained with H&E to examine immune infiltration or stained with LFB to determine the extent of demyelination. At least three mice per group were used for this study. Lesions were identified by dotted lines in the LFB stained sections, and data were collected using Image J. For immunostaining, spinal cords were frozen in optimal cutting temperature mounting medium, and sectioned at 20 $\mu$m. Sections were blocked in 5% donkey serum in PBS with 0.03% tween 20, then incubated with primary antibodies (MBP 1:500; Iba1 1:200 diluted with blocking buffer) at 4°C overnight, followed by 1 h incubation at RT with anti-rabbit IgG or anti-mouse IgG fluorescent conjugated secondary antibody. Nuclei were stained with DAPI (Invitrogen) and mounted with Vectashield mounting medium (Vector Laboratories).

## Toluidine blue staining

To assess the amount of myelinated axons, EAE mice were perfused with 4% paraformaldehyde and 2.5% glutaraldehyde in sodium cacodylate (Electron Microscopy Sciences). Spinal cord sections were post-fixed with 1% $OsO_4$ for 4–6 h, dehydrated, and embedded in epoxy resin. Samples were sectioned at 500 nM and stained with 1% Toluidine Blue. Samples were examined by light microscopy, and the percentage of myelinated fibers was calculated as a percentage of the total number of axons.

## PEMs isolation

Mice were injected with 3% thioglycollate solution intraperitoneally, and 4 d later macrophages were collected by peritoneal lavage. The

cells were plated on culture dishes, followed by medium change the next day. Macrophages were isolated by their ability to remain attached to the culture dishes. The cells were either treated with 1 $\mu$g/ml LPS to stimulate cytokine secretion or were left untreated. RNA was isolated by RNeasy kit and subjected to qPCR using SYBR green. The supernatants were analyzed by CCL2 ELISA kit (eBioscience) according to the manufacturer's instructions.

## Proliferation assays and cytokine analysis

Mice were immunized subcutaneously with MOG/CFA at the base of the tail or on the back, and lymph nodes or spleens were collected 11 d later. Single cell suspensions were prepared and 4 × 10⁵ cells were cultured in triplicates in 96-well plates in the presence of increasing concentrations of MOG peptide. 48 h later, the supernatants were harvested and the cultures pulsed with 1 $\mu$Ci of ³H-thymidine for an additional 14 h. The supernatants were then spun down and stored at –80°C until analysis. Quantification of cytokines in the supernatant was performed by Quansys Biosciences.

## mRNA analysis

Total RNA was isolated from tissues or cells with TRIzol reagent (Invitrogen). cDNA was synthesized from 1 $\mu$g of RNA using the qScript cDNA synthesis kit (Quanta Biosciences). qRT–PCR was performed using iQ SYBR green supermix (Bio-Rad) with gene-specific primers (Table S1). Gene expression was normalized to the housekeeping gene hypoxanthine guanine phosphoribosyl transferase (HPRT) and GAPDH. Fold change in gene expression was analyzed by the $2^{-(\Delta\Delta CT)}$ method.

## Flow cytometric analysis

For peripheral immune analyses, mice were injected with MOG/CFA emulsion (Hooke Labs) at the base of their tails. 12 d post injection, spleen and lymph nodes were harvested and cultured with 20 $\mu$g MOG for 48 h and stained for surface markers using perCP-Cy5.5 conjugated anti-CD11b, FITC conjugated anti-CD11c, Pacific blue conjugated anti-CD4, APC conjugated anti-CD8, and Alexa Fluor 700 anti-CD3 antibodies. Live dead yellow (Invitrogen) was used to analyze live populations. For intracellular cytokine staining, during the last 4 h of culture, PMA (50 ng/ml), ionomycin (500 ng/ml), and Golgi plug were added to the culture. Cells were first stained with surface markers, CD3, CD4, and CD8, fixed, and permeabilized with Cytofix/Cytoperm solution according to the manufacturer's instructions (BD Biosciences) and then stained intracellularly with PE conjugated anti-IFN-γ. For isolation of mononuclear cells from spinal cords, the lumbar regions of the spinal cords were isolated from PBS-perfused EAE mice post immunization. Tissues were disrupted using a Dounce homogenizer and were fractionated on a 70–30% Percoll gradient. Mononuclear cells were recovered from the 70–30% interphase, and cells were stained for flow cytometry analysis.

## In vitro migration assay

Migration of thioglycollate-elicited PEMs was measured in a modified Boyden chamber using Transwell inserts with a 5 $\mu$m porous

membrane (Corning). PEMs from WT mice were loaded in the Transwell chamber and conditioned medium, diluted 1:2, and placed in the lower chamber. Conditioned medium was obtained by isolating splenocytes from WT or HDAC11 KO mice post MOG immunization and culturing them with MOG peptide for 48 h. The medium was collected, centrifuged, and used as a chemoattractant for WT PEMs. After 3 and 24 h, cells on the upper side of the membranes were removed while the nuclei of migratory cells on the lower side of the membrane were stained with DAPI. The number of migratory cells was counted with a fluorescence microscope.

### Lentivirus infection

The lentivirus transduction particles containing shRNA, specific for HDAC11 (SHCLNG-NM_024827) or non-targeting shRNA (SHC002V), were purchased from Sigma-Aldrich. The lentivirus was prepared by transfecting HEK 293 cells with the destination plasmid and packaging mix (Life Technologies), as described (Kutner et al, 2009). Cell culture supernatant containing the lentivirus was harvested 60 h after transfection, filtered through 0.45 $\mu$m filters, and mixed with polybrene (10 $\mu$g/ml; Santa Cruz Biotechnology) and used to transduce U251 cells. 3 d after transduction, puromycin was added to select a stable expression cell line.

### Luciferase reporter assay

Firefly and Renilla luciferase activity were measured in cell lysates using a Dual Luciferase Reporter Assay System (Promega) following the manufacturer's protocol. The data were expressed as a ratio of Firefly to Renilla luciferase activity to normalize for transfection efficiency. Experiments were performed at least three times.

### Immunoprecipitation assay

HEK 293 cells were transfected with plasmids, and 48 h later, cell lysates were prepared in EBC buffer (50 mM Tris-HCl [pH 8.0], 120 mM NaCl, 0.5% NP-40, 10 $\mu$g of aprotinin per ml, 10 $\mu$g of leupeptin per ml, 0.1 mM phenylmethylsulfonyl fluoride, 50 mM NaF, 1 mM sodium orthovanadate, 1 mM EDTA) as described previously (Dalal et al, 1999). Cleared extracts were incubated with anti-Flag or anti-PU.1 antibody overnight at 4°C. Ten $\mu$l of precleared protein A/G beads were subsequently added and incubated for 2 h. The immune complexes were washed three times with NET-N buffer (10 mM Tris, pH 8.0, 150 mM NaCl, 1mM EDTA, 1% NP-40), followed by Western blot analysis.

### ChIP assay

PEMs were crosslinked with 3'-dithiobispropionimidate (Pierce Biotechnology) for 30 min. The cells were then washed with PBS and incubated with Tris-solution (100 mM Tris, pH 8.0, 150 mM NaCl). The cells were crosslinked with 1% formaldehyde at RT. The crosslinking reaction was quenched with 0.125 M glycine at RT for 5 min. The cells were then washed with cold PBS, and resuspended in TX-100/NP40 buffer (10 mM Tris, pH 8, 10 mM EDTA, 0.5 M EGTA, 0.25% TX-100 and 0.5% NP-40, and protease inhibitor). The cells were resuspended in

ice cold salt wash buffer (10 mM Tris, pH 8, 1 mM EDTA, 0.5 M EGTA, 200 mM NaCl, and protease inhibitors), incubated for 10 min, and lysed in sonication buffer (10 mM Tris, pH 8, 1 mM EDTA, 0.5 M EGTA, 1% SDS at a cell density of 1 × 10$^6$ cells/30 $\mu$l, and protease inhibitors). Sonication was performed six times for 10 s each to produce chromatin fragment lengths of 200 to 1,000 bps. The lysates were centrifuged at 12,000 $g$ at 4°C for 10 min. The supernatants were diluted fivefold with ChIP dilution buffer (50 mM Tris–HCl, pH 8.0, 167 mM NaCl, 1.1% Tx-100, and 0.11% sodium deoxycholate). Aliquots of the lysates were stored as input DNA. The remainder of the supernatants were incubated overnight at 4°C with anti-PU.1 or the respective control IgG. Immune complexes were precipitated with protein A or G (Invitrogen) and, subsequently, the beads were washed successively with low-salt buffer (50 mM Tris–HCl, pH 8.0,150 mM NaCl, 1 mM EDTA, 0.1% SDS, 1% Tx-100, and 0.1% sodium deoxycholate). The DNA samples were uncrosslinked and incubated with proteinase K at 55°C for 2 h and purified using the Qiagen DNA extraction kit. The DNA was used for qPCR to test the occupancy of the protein at the genomic locus.

## Supplementary Information

## Acknowledgements

We thank Thomas Rosahl at Merck for the *Hdac11* constitutive KO mouse line, which was generated at the Institut Clinique de la Souris and obtained via Taconic. We also thank Nabeel R Yaseen at Washington University in St. Louis for the PU.1 deletion constructs. In addition, we are immensely grateful to Sonali Bahl, Zahra Shokatian, and Raneen Rahhal for their comments that greatly improved the manuscript. This research was supported by grants from the National Institutes of Health. Some preliminary experiments described in this paper were performed at the Moffitt Cancer Center.

### Author Contributions

L Sun: data curation, investigation, methodology, and writing—review and editing.

E Telles: conceptualization, data curation, investigation, methodology, and writing—review and editing.

M Karl: data curation, investigation, methodology, and writing—review and editing.

F Cheng: investigation and methodology.

N Luetteke: conceptualization, data curation, investigation, methodology, and writing—review and editing.

EM Sotomayor: investigation, methodology, and writing—review and editing.

RH Miller: resources, data curation, formal analysis, investigation, methodology, and writing—review and editing.

E Seto: conceptualization, resources, data curation, formal analysis, funding acquisition, validation, investigation, methodology, project administration, and writing—original draft, review, and editing.

## Conflict of Interest Statement

The authors declare that they have no conflict of interest.

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
