## [Reviewer comments · Life Science Alliance]

Life Science Alliance

Loss of HDAC11 ameliorates clinical symptoms in a multiple sclerosis mouse model

Lei Sun, Elphine Telles, Molly Karl, Fengdong Cheng, Noreen Luetke, Eduardo Sotomayor, Robert Miller, and Edward Seto

DOI: 10.26508/lsa.201800039

Corresponding author(s): Edward Seto, George Washington University

Review Timeline:

Submission Date:	2018-02-23
Editorial Decision:	2018-04-05
Revision Received:	2018-08-18
Editorial Decision:	2018-09-11
Revision Received:	2018-09-16
Accepted:	2018-09-17

Scientific Editor: Andrea Leibfried

Transaction Report:

April 5, 2018

Re: Life Science Alliance manuscript #LSA-2018-00039-T

Prof. Edward Seto
George Washington University
GW Cancer Center
800 22nd St NW
Room 8800
WASHINGTON, DC - District Of Columbia 20052

Dear Dr. Seto,

Thank you for submitting your manuscript entitled "Loss of HDAC11 ameliorates clinical symptoms in a multiple sclerosis mouse model" to Life Science Alliance. The manuscript was assessed by expert reviewers, whose comments are appended to this letter.

As you will see, the reviewers appreciate your work. However, they also note various issues that currently preclude publication here. The proposed mechanisms regarding CCL2 transcription is not sufficiently supported by the data provided and is also at odds with the known literature. Furthermore, the referees think that more insight is needed to better support your conclusions.

I would like to invite you to revise your manuscript following the constructive input the reviewers provide. Please note that I will need strong support from the reviewers on such a revised version. Importantly, for publication the following issues of the reviewers need to be addressed:

- investigate degree of neurodegeneration
- analyze whether HDAC11 protein per se or the catalytic activity is regulating CCL2 expression, and add discussion on the CLL2 data and de-emphasize this part
- provide a more general analysis of the KO mice
- discuss alternative explanations for the protection observed
- provide a robust statistical analysis

-- High-resolution figure, supplementary figure and video files uploaded as individual files: See our detailed guidelines for preparing your production-ready images, <http://life-science-alliance.org/authorguide>

B. MANUSCRIPT ORGANIZATION AND FORMATTING:

Full guidelines are available on our Instructions for Authors page, <http://life-science-alliance.org/authorguide>

Thank you for this interesting contribution to Life Science Alliance. We are looking forward to receiving your revised manuscript.

Sincerely,

Reviewer #1 (Comments to the Authors (Required)):

This is an interesting study on the role of an enigmatic HDAC in an autoimmune demyelinating disorder. Although the whole study may be convincing, the authors are encouraged to:

- 1-explain why HDAC11 KO mice have a similar score in the first phase of the disease.
- 2-in the second phase symptoms are slightly reduced but demyelination is much more robustly decreased.
- 3-Investigate the degree of neurodegeneration
- 4-explain why a transcriptional inhibitor such as HDAC11 should promote CCL2 transcription
- 5-why loss of HDAC11 affords protection in the demyelinating models with cuprizone or lysophosphatidyl-choline considering that the latter are not due to immune infiltration.
- 6-whether HDAC11 protein per se or its activity regulate CCL2 expression. this is of key relevance given that the authors claim that selective chemical inhibitors might be of therapeutic relevance to MS.
- 7-why CD4+ cells infiltration is not altered in the KO mice

Reviewer #2 (Comments to the Authors (Required)):

Sun et al study the role of HDAC11 in EAE, a model for multiple sclerosis. By generating HDAC11 ko mice they show that loss of HDAC11 ameliorates phenotypes specifically associated with the late phase of the disease. Mechanistically the authors provide evidence that HDAC11 regulates the expression of CCL2 via its interaction with the microglia transcription factor PU1. The authors provide a lot of data that is very interesting and timely. To decipher the potential of specific HDAC for the treatment of MS is certainly of great importance. Therefore the data is very interesting and suggests that HDAC11 inhibitors could be suitable to treat MS. I only have a few comments.

1.

I understand that the HDAC11 ko mice were generated in collaboration with Taconic and represent a full knock out from early developmental stages. The authors show that there are no compensatory expression changes of other HDACs but since this appears to be the first report on these mutant mice, some more basal analysis would be necessary. For example, body weight, brain weight, brain anatomy, overall status of organs, life span, etc.

2.

Related to point 1 I think it's a bit strange that HDAC11 KO and WT mice are only investigated in response to EAE. I believe an important control group would be sham-treated WT and HDAC11 ko mice. Maybe such data could be added at least on part.

3.

The authors say that "For example, broad spectrum HDACi, Vorinostat, trichostatin A (TSA), and valproic acid have been tested for their efficacy in EAE mice and have been shown to ameliorate EAE.. However, these inhibitors are not specific in their mode of action, resulting in suboptimal therapeutic outcomes, and unwanted serious adverse effects". While this is true for TSA, Vorinostat is known to show some selectivity for especially class I HDACs and HDAC6. Moreover

Vorinostat and Valporic acid are approved drugs that re given to humans. Thus, the authors statement should be reworded.

Reviewer #3 (Comments to the Authors (Required)):

The article by Sun and colleagues investigates the role of HDAC11 in experimental autoimmune encephalomyelitis (EAE), a mouse model for multiple sclerosis (MS). The experiments presented in the article show that the loss of HDAC11 in KO mice reduces the demyelination of the spinal cord and ameliorate some pathological traits. Their results suggest that the lack of HDAC11 prevents the infiltration of immune cell into the CNS through the control of the expression of the pro-inflammatory chemokine C-C motif ligand 2 (CCL2) directed by the PU.1 transcription factor. According to the authors, these results underscore the potential use of HDAC11 inhibitors for the treatment of MS.

Specific comments:

1. The repression of anti-inflammatory responses by HDAC11 has been described before (see recent review on this topic by Yanginlar and Logie, 2018). Therefore, the novelty of this study is limited. The authors focused in CCL2 as a possible target but other important molecules involved in inflammation have been previously identified as targets of HDAC11, such as interleukin IL10. Is the mechanism of action proposed, involving PU.1, exclusive of CCL2?, or does it have more general implications in the regulation of anti-inflammatory responses, affecting several related targets?
2. The presentation of statistics should be consistent and clear in text, figures and figure legends. Very frequently the authors refer to "significant" but they do not provide any statistical parameters. The use of asterisks or other symbols to represent p-values below a given threshold should be added to figures and sample size in each experimental condition should be included in the legends. Surprisingly, several figure legends refer to p-values but these are not included either numerically or with symbols in the figures. For example, in the description of Figure 5 the authors refer to some significant differences that are difficult to appreciate in the figures. The authors should revise ALL figures to clearly indicate the significance of all relevant comparisons.
3. The model proposed by the authors seems to contradict more of what we know about transcription factors and their interaction with epigenetic enzymes. The common view is that transcription factors (by definition) contain a DNA binding domain that recognize and bind specific sequences in the DNA; they also act as an anchor for epigenetic enzymes that are recruited by the transcription factor to specific sites in the genome. These enzymes usually lack a DNA binding domain and indirectly interact with DNA. The authors propose just the opposite: HDAC11 recruits PU.1 to the DNA. How do they envision this recruitment? Does HDAC11 bind and recognize specific sequences in the DNA (as far as I know, it does not)? One possibility that could explain the results presented in figure 7 is that the DNA binding ability of PU.1 depends on its acetylation state. Is PU.1 a direct substrate of HDAC11? This possibility that could conciliate the authors' result with the common model should be explored and discussed in the paper. For example, the authors could evaluate the efficacy of a catalytic-dead, but otherwise complete HDAC11 protein, in the assays presented in figures 6 and 7.
4. Less surprising, but also worth discussing is the fact that they are proposing that a protein generally associated with transcription repression seems to activate the expression of CCL2. Do

the authors propose a direct mechanism for the activation? Does this mechanism relate to histone acetylation or non-histone substrates?

Minor:

5. The discussion could be easily shortened.

6. Page 17: The article has 7 figures and 5 supplemental figures; there is no reason to indicate "data not show". This gene expression result could be easily accommodated in one of the figures.

7. Page 17: The authors indicate that "HDACis have a long history of use in psychiatry and neurology as mood stabilizers and anti-epileptics". This is not accurate. They are referring exclusively to valproic acid, a compound with activity as HDACi but not only as HDACi. The value as mood stabilizer and anti-epileptic has not been extended to other HDACis.

8. Figure 2A: Could the authors quantify the difference?

9. Figure 6: The presentation of these results is unnecessarily complicated and confusing. Please revise. What is presented in the two upper blots in Figure 6A? The legend indicates: "Two different representative exposures of PU.1 IB are shown." Does this mean two independent experiments or two blots of the same samples? What are the extra bands recognized by the anti-PU.1 antibody in one of the blots? Molecular weight markers should be added to the panels along with information regarding the expected molecular weights of the truncated proteins.

Reviewer #1

We thank Reviewer #1's comment that "This is an interesting study... the whole study may be convincing..." Our point-by-point response to the reviewer's comment:

1. "The authors are encouraged to explain why HDAC11 KO mice have a similar score in the first phase of the disease."

One of the key discoveries of our study is that HDAC11 regulates the clinical outcomes of EAE mainly through CCL2 expression. Our observations that HDAC11 KO mice have similar scores in the early phase of the disease are in line with previous reports that anti-CCL2 treatment reduces CNS macrophage accumulation during the relapsing phase, but not in the acute phase of the disease (J Neuroimmunol 1998; 92: 98-108). Also, because CCL2 plays a minimum role in patients with the early active phase of MS (Eur J Neurol 2004;11:445-449), our result that HDAC11 KO attenuated symptoms in chronic, but not the early phase of, EAE is again consistent. We have now incorporated these points in our manuscript.

2. "In the second phase, symptoms are slightly reduced but demyelination is much more robustly decreased."

As expected, the EAE clinical scores (symptoms) and demyelination lesions determined by LFB staining is qualitatively consistent, but not strictly quantitatively proportional. EAE is a multifocal and random disease, such that it is difficult to predict where lesions will occur, especially when the total lesion volume of the spinal cord is reduced. LFB stained spinal cords only show severe demyelinated lesions, but not minor injuries and reduction of preserved axons. In contrast, toluidine blue staining and electron microscope images better show the ultrastructure of the myelinated axon. Therefore, we now added the results of toluidine blue staining in Figures 1D and 1E. These new results further confirm that, in the chronic phase of EAE, the number of preserved axons in the lesion area is significantly higher in HDAC11 KO compared to WT mice and that HDAC11 KO promotes recovery via remyelination in the chronic phase of EAE. The results also suggest that, perhaps in the chronic phase of EAE, some mild lesions with more preserved axons could not be identified using LFB staining.

3. "Investigate the degree of neurodegeneration."

We compared the number of motor neurons in the grey matter of EAE mouse spinal cords with toluidine blue stain, and also stained non-phosphorylated neurofilament H with anti-SMI-32 antibody. These new results show no obvious difference between WT and HDAC11 KO suggesting that neurodegeneration is not a major cause of reduced clinical symptoms in the chronic phase of EAE in HDAC11 KO mice. We have now added these results in Figure S4.

4. "Explain why a transcriptional inhibitor such as HDAC11 should promote CCL2 transcription."

It is a common misconception that the chief function of HDACs is to serve as transcriptional repressors (inhibitors). Just as one example, although HDAC3 represses transcription when targeted to promoters and serves as a corepressor (J Biol Chem 1997; 272:28001-28007), HDAC3 also is required for transcriptional activation (Cell 2000; 102:753-763). In cells derived from Hdac3 knockout mice, both up-regulation and down-regulation of gene expression were detected (Mol Cell 2008; 30:61-72). Also, in gene expression profiling studies comparing cells treated and not treated with HDAC inhibitors, the number of genes

down-regulated was comparable to the number of up-regulated genes (e.g., *BMC Med Genomics* 2009; 2:67). Like HDAC3 (and maybe all HDACs), it is not surprising that HDAC11 can also activate transcription. In this case, the mechanism is by altering the recruitment of PU.1 (Figures 6 and 7). Furthermore, recent reports indicate that HDAC11 is a fatty-acid deacylase rather than a histone deacetylase (*ACS Chem Biol* 2018; 13:685-693; *Cell Chem Biol* 2018; 25:849-856). We predict that HDAC11 regulates the expression of CCL2, not through histone deacetylation, but by affecting the binding of the transcriptional factor PU.1 to the promoter of CCL2.

5. "Why loss of HDAC11 affords protection in the demyelinating models with cuprizone or lysophosphatidyl-choline considering that the latter are not due to immune infiltration."

The reviewer is correct that demyelination induced by cuprizone and lysophosphatidyl-choline (LPC) are indeed not due to immune infiltration. In the cuprizone model experiments, the rate and extent of demyelination between WT and HDAC11 KO are very similar. Based on these results, we in fact cannot conclude at this time that the loss of HDAC11 affords protection in the demyelinating models with cuprizone. Likewise, in the LPC model, though there is a modest reduction in the size of spinal cord lesions in HDAC11 KO mice compared to WT mice 15 days after injection, there is no statistically significant difference between the two groups. Therefore, we conclude that the role of HDAC11 in EAE most likely involves immune responses, although we cannot completely rule out the possibility of an additional non-immune mediated component.

6. "...whether HDAC11 protein per se or its activity regulate CCL2 expression."

We now generated a HDAC11 H143A mutant, which is catalytically-dead in its defatty-acylation and deacetylase activities (*ACS Chem Biol* 2018; 13:685-693 and *Mol Cells* 2017; 40:667-676). This mutant did affect the interaction of HDAC11 with PU.1 and regulation of CCL2 expression (Figures S8B and S8C). These new results suggest that the enzymatic activity of HDAC11 is important for regulating CCL2 expression. These new observations are encouraging because they confirm that selective HDAC inhibitors might be of therapeutic relevance to MS. Nevertheless, it is important to point out that currently it is technically challenging to determine if HDAC11 protein per se or its activity regulates CCL2 expression. It is beyond the scope of this paper and frankly, quite difficult, to produce a HDAC11 mutant that would affect the activity of HDAC11 alone without affecting the protein.

7. "Why CD4+ cells infiltration is not altered in the KO mice?"

In the EAE model, T cells play a central role in directing the immune response, and both CD4+ and CD8+ T cells are important in MS. But a recent study points out that "Although the majority of research on MS pathogenesis has centered on the role of effector CD4 T cells, accumulating data suggests that CD8 T cells may play a significant role in the human disease. In fact, in contrast to most animal models, the primary T cell found in the CNS in patients with MS, is the CD8 T cell. As patient-derived effector T cells are also resistant to mechanisms of dominant tolerance such as that induced by interaction with regulatory T cells (Tregs), their reduced response to regulation may also contribute to the unchecked effector T-cell activity in patients with MS (*Cold Spring Harb Perspect Med* 2018; 8(4) pii: a029025)." CD4+ cells include Treg cells, which maintain tolerance to self-antigens and prevent autoimmune disease. It has been reported that HDAC11 deletion promotes Foxp3+ Treg function, and led to long-term survival of fully MHC-mismatched cardiac allografts (*Sci Rep* 2017; 17:8626). So maybe CD4+ cells include more Treg cells in HDAC11 KO mice, and the

total amount of CD4+ cells has little change. We have modified our discussion to reflect this point.

Reviewer #2

We thank Reviewer #2's comment that "The authors provide a lot of data that is very interesting and timely... is certainly of great importance... the data is very interesting..." Our point-by-point response to the reviewer's comment:

1. In the HDAC11 KO mice, "The authors show that there are no compensatory expression changes of other HDACs but since this appears to be the first report on these mutant mice, some more basal analysis would be necessary."

We thank the reviewer for this valuable suggestion. We have now compared the body weights, brain weights and brain anatomy between WT and HDAC11 KO female mice at age 20-weeks. These new results, presented in Figure S3, did not show an obvious difference between the two groups. Further, compared to WT mice, HDAC11 KO mice were healthy and showed no apparent abnormal appearance. The lifespans of both WT and HDAC11 KO mice exceeded 1 year.

2. "An important control group would be sham-treated WT and HDAC11 KO mice."

Our HDAC11 KO and WT mice are on C57BL/6 background. It has already been reported that in C57BL/6 mice, the EAE clinical score of sham-treated groups are always zero (e.g., J Neuroimmunol 2017; 310:51-59 and Clin Sci 2015; 128:95-109). We do not think it is necessary to repeat these well-documented results.

3. "Broad spectrum HDACi... have been tested for their efficacy in EAE mice... However, these inhibitors are not specific... While this is true for TSA, Vorinostat is known to show some selectivity for especially Class I HDACs and HDAC6. Moreover, Vorinostat and Valproic acid are approved drugs... Thus, the authors' statement should be reworded."

In accordance with the reviewer's suggestion, we have modified this statement.

Reviewer #3

Our point-by-point response to the reviewer's comment:

1. "The authors focused in CCL2 as a possible target but other important molecules involved in inflammation have been previously identified as targets of HDAC11, such as interleukin IL10. Is the mechanism of action proposed, involving PU.1, exclusive of CCL2?, or does it have more general implications in the regulation of anti-inflammatory responses, affecting several related targets?"

We agree with the reviewer that there are other HDAC11 targets in addition to CCL2. We assayed many cytokine/chemokine levels, including IL-10, in HDAC11 WT and KO mouse splenocytes (Figure 4B). Because in our current model, CCL2 showed the greatest change while IL-10 had no significant change, the logical choice for us is to focus on CCL2. This does not mean that IL-10 is not important as a HDAC11 target in other systems. Rather, for our current study of the role of HDAC11 in EAE, CCL2 is a key molecule. We do not believe the mechanism of action proposed, involving PU.1, is exclusive of CCL2. For example, it has

been reported that PU.1 transcriptionally regulates CD11b, which is selectively expressed in mature monocytes, macrophages, granulocytes and natural killer cells (J Biol Chem 1993; 268:5014-5020). Our data (Figure 3E) indicating that, the CD11b mRNA level in the HDAC11 KO mouse spinal cords was decreased compared to WT are consistent with these early findings.

2. "The presentation of statistics should be consistent and clear in text, figures and figure legends..."

We greatly appreciate the reviewer's suggestion on this. We have now added statistical parameters in the figures and descriptions in the figure legends.

3. "The model proposed by the authors seems to contradict more of what we know about transcription factors and their interaction with epigenetic enzymes... Does HDAC11 bind and recognize specific sequences in the DNA (as far as I know, it does not)? Is PU.1 a direct substrate of HDAC11...? The authors could evaluate the efficacy of a catalytic-dead, but otherwise complete HDAC11 protein..."

There is now abundant evidence from different groups that HDAC11 is not a typical epigenetic enzyme. We agree with the reviewer that HDAC11 does not bind and recognize specific DNA sequences. Consistent with this notion, in a ChIP assay we found less PU.1 binding to the CCL2 promoter in HDAC11 KO compared to WT cells, suggesting that HDAC11 regulates PU.1 recruitment to the CCL2 promoter. PU.1 recognizes specific promoter sequences and binds to DNA with its ETS domain, and we found that HDAC11 could bind to the ETS DNA binding domain of PU.1, consequently affecting the interaction of PU.1 with DNA.

We have tested the acetylation level of PU.1 with an anti-acetylated-lysine antibody in the presence and absence of HDAC11, but did not detect acetylated PU.1. Our data is consistent with recent reports that HDAC11 deacetylase activity is extremely low and, therefore, PU.1 may not be a direct deacetylation substrate of HDAC11. We have decided not to include these negative data in the current paper.

We thank the reviewer's suggestion to evaluate the efficacy of a catalytic-dead HDAC11. The results are now shown in Figures S8B and S8C.

4. "Less surprising, but also worth discussing is the fact that a protein generally associated with transcription repression seems to activate the expression of CCL2..."

It is incorrect to believe that HDAC11 is generally associated with transcription repression. There are reports that HDAC11 represses transcription when those experiments were designed to examine repression. Like most HDACs, our data suggest HDAC11 can also activate transcription. However, we do not think HDAC11 directly activates the transcription of CCL2. Rather, through binding to the ETS domain of PU.1, HDAC11 affects the interaction of PU.1 with the CCL2 promoter. This mechanism is most likely not involved with histone acetylation or non-histone substrates. We have included these points in the Results and Discussion sections.

5. The discussion could be easily shortened.

We have done our best to shorten the discussion without losing clarity in the manuscript. If necessary, we will trim the manuscript further to conform to the journal style.

6. "There is no reason to indicate 'data not shown.' This gene expression result could be easily accommodated in one of the figures."

We agree. We have now added these results to Figure S8A.

7. The statement that "HDACs have a long history of use in psychiatry and neurology as mood stabilizers and anti-epileptics is not accurate..."

We appreciate the reviewer for pointing this out to us. We have now deleted this sentence from our paper.

8. "Figure 2A: Could the authors quantify the difference?"

We again appreciate the reviewer's suggestion and have now added quantification results in Figure 2C.

9. "Figure 6: The presentation of these results is unnecessarily complicated and confusing..."

We have now modified Figure 6 by deleting the lighter exposure blot in Figure 6A, and added molecular weight markers to the panels. We thank the reviewer for these suggestions.

September 11, 2018

RE: Life Science Alliance Manuscript #LSA-2018-00039-TR

Prof. Edward Seto
George Washington University
GW Cancer Center
800 22nd St NW
Room 8800
Washington DC, DC 20052

Dear Dr. Seto,

Thank you for submitting your revised manuscript entitled "Loss of HDAC11 ameliorates clinical symptoms in a multiple sclerosis mouse model". Your revised work has now been evaluated by the original reviewers again.

As you can see below, while reviewer #2 now supports publication, reviewer #1 and #3 still raise some issues. We would therefore like to ask you to respond to the remaining concerns and to provide a final version of your manuscript. We realize that reviewer #1 raises new points (point 2 and 4), and we don't expect that you address these with additional experiments but rather with a balanced discussion. All other points should get addressed as well (by discussion/re-arrangement of the data already at hand). Additionally, please pay attention to the following editorial points:

- please provide less over-contrasted blots for figure 6
- please provide the supplementary figure files without legends, the legends should get incorporated into the main manuscript text file.

A. FINAL FILES:

-- High-resolution figure, supplementary figure and video files uploaded as individual files: See our detailed guidelines for preparing your production-ready images, <http://life-science-alliance.org/authorguide>

-- Summary blurb (enter in submission system): A short text summarizing in a single sentence the study (max. 200 characters including spaces). This text is used in conjunction with the titles of

papers, hence should be informative and complementary to the title. It should describe the context and significance of the findings for a general readership; it should be written in the present tense and refer to the work in the third person. Author names should not be mentioned.

B. MANUSCRIPT ORGANIZATION AND FORMATTING:

Full guidelines are available on our Instructions for Authors page, <http://life-science-alliance.org/authorguide>

Sincerely,

Andrea Leibfried, PhD
Executive Editor
Life Science Alliance
Meyershofstr. 1
69117 Heidelberg, Germany
t +49 6221 8891 502
e a.leibfried@life-science-alliance.org
www.life-science-alliance.org

Reviewer #1 (Comments to the Authors (Required)):

This is an interesting and timely study on the role of HDAC11 in a model of MS. Still, in this reviewer's opinion the study appears preliminary and additional experiments should be performed to provide the reader with a detailed picture of the role of HDAC11 in the autoimmune response to the CNS.

1- the ability of HDAC11 suppression to promote remyelination without targeting neurodegeneration is not convincing. As shown in Fig. 1D, in the spinal cord of KO mice at day18 a widespread axonal loss is present (higher magnification of this section would help to better appreciate the degree of degeneration). Of course this is in contrast with the apparent preservation of axonal structures at day 36. Also, the absence of neurodegeneration is at odds with the authors' claim that this is a model of progressive MS.

2- The impact of HDAC11 suppression on Treg dynamics and function in the EAE model should be investigated, also in light of their recent study on this subject.

3- the key point of the study is the identification of HDAC11 as a potential target for MS therapy. The authors however should soften the statement given that their results are related to a model of genetic suppression that is dramatically different from a model of acute pharmacological inhibition. In this light, the use of the HDAC11 inhibitors used by the authors in a prior study (Sci Rep 2017) is encouraged.

4- The authors state that HDAC11 mainly regulates EAE development by suppressing CCL2 expression. In this study a causal relationship between chemokine suppression and clinical amelioration is lacking. Importantly, why CCL2 reduction was more pronounced in the spinal cord at day 40 than at day 19? The authors should discuss the complex phenotype of HDAC11 KO mice as emerges by their recent contributions. In this light several mechanisms might have contributed to reduced EAE symptoms.

5-Fig. 2C. Nuclear staining demonstrates severe loss of cellularity even though myelin content (or neurofilament, Fig S4B) is not reduced. The authors should comment on this apparent discrepancy.

Reviewer #2 (Comments to the Authors (Required)):

the authors have addresses all previous concerns.

I suggest publication of the study

Reviewer #3 (Comments to the Authors (Required)):

The revised article includes new experiments and adds new controls and analyses. Most of my concerns have been addressed. However, some minor revisions are still necessary.

Main (related to point 3 in my first report):

The authors indicate in page 13 that they "hypothesized that HDAC11 recruits PU.1 to the promoter of the CCL2 gene". Consistent with the authors' response to my comment ("we found that HDAC11 could bind to the ETS DNA binding domain of PU.1, consequently affecting the interaction of PU.1 with DNA"), they should rephrase that sentence. For example, "we hypothesized that HDAC11 enables the binding of PU.1 to the promoter of the CCL2 gene" or "we hypothesized that HDAC11 affects the interaction of PU.1 with DNA at the promoter of the CCL2 gene".

The also indicate in the Abstract that HDAC11 regulates CCL2 by "binding to the ETS domain of the PU.1 transcription factor and recruiting it to the CCL2 promoter". Their results do not show that

HDAC11 recruits PU.1 to the promoter; instead they show that HDAC11 is required for the binding of PU.1 to that promoter. Different mechanisms, direct and indirect, could explain that requirement. The authors should discuss in the manuscript how do they envision this regulatory mechanism. In the rebuttal letter they wrote that "recent reports indicate that HDAC11 is a fatty-acid deacylase rather than a histone deacetylase" and predict that "HDAC11 regulates the expression of CCL2, not through histone deacetylation, but by affecting the binding of the transcriptional factor PU.1 to the promoter of CCL2" (a model that is compatible with the histone-independent effect observed in reporter plasmid assays). Are they proposing that fatty-acid deacylation indirectly leads to changes in PU.1 binding? Another option is that PU.1 binding was directly regulated by acetylation. The authors indicate in the rebuttal that they failed to detect changes in the acetylation level of PU.1, but this negative result does not discard that possibility (the antibody used could not be adequate). Other options could involve the acetylation of NF- κ B (with binding to sites next to those for PU.1) that somehow recruits PU.1, etc (since NF- κ B binds specific DNA sequences the use of the term "recruitment" seems more appropriate in that case).

- The assessment of a catalytic-dead HDAC11 represents an important control. I suggest to move the new panels in Figure S8 to Figure 7.

- The authors indicate that the new panel S8A shows that there is no change in PU.1 expression upon overexpression or KO of HDAC11. However only the overexpression condition is shown. Also the panels S8B and S8C are introduced much earlier than panel S8A.

Reviewer #1

We thank Reviewer #1's comment that "This is an interesting study and timely study on the role of HDAC11 in a model of MS." Our point-by-point response to the reviewer's comments:

1. "The ability of HDAC11 suppression to promote remyelination without targeting neurodegeneration is not convincing... Also, the absence of neurodegeneration is at odds with the authors' claim that this is a model of progressive MS."

We appreciate the reviewer pointing out, in the previous as well as in the second review, that neurodegeneration may be a contributor to clinical outcomes in progressive MS. In Figure 1D, E, preserved axon reduction appears most significantly in the lesion area, but is not widespread over all white matter. We agree that understanding neurodegeneration is important to elucidate how to treat progressive MS. We are also aware of debates regarding whether neurodegeneration is an independent process in patients with MS or its occurrence is secondary to inflammation. As with any research models, there are limitations to the EAE model for MS research and we acknowledge that it may not fully address the role of HDAC11 in neurodegeneration. However, the goal of this paper is not to identify sensitive measures of neurodegeneration in HDAC11 WT and KO mice that will be suitable for use as outcome measures in experimental therapeutics. In contrast to the reviewer's comment, we do not claim that there is an "absence of neurodegeneration." Rather, our data so far support the hypothesis that neurodegeneration is not the major cause of reduced clinical symptoms in the chronic phase of EAE in our HDAC11 KO mice. We have modified our discussion to clarify this point.

2. "The impact of HDAC11 suppression on Treg dynamics and function in the EAE model should be investigated..."

We completely agree with the editor that the study of Treg in EAE is beyond the scope of the current manuscript and detracts from the focus of our paper. The paper that HDAC11 targeting promotes Foxp3+ Treg function is cited in the current manuscript.

3. "The authors should soften the statement given that their results are related to a model of genetic suppression that is dramatically different from a model of acute pharmacological inhibition... the use of the HDAC11 inhibitors is encouraged."

We agree that genetic suppression and acute pharmacological inhibition are different models, and sometimes they reveal different phenotypes. However, we disagree with the reviewer that we need to soften our conclusion in this current study. As pointed out in our discussion, the course of disease progression observed in HDAC11 KO mice with EAE (genetic suppression) fits well with a previous study of HDAC inhibitor-treated EAE mice (pharmacological inhibition), wherein clinical amelioration was observed only during the chronic progressive phase (Camelo et al. 2005). In this case, our genetic suppression data are not "dramatically different," but rather in perfect harmony with a model of acute pharmacological inhibition.

The reviewer's suggestion that we explore the use of HDAC11 inhibitors is a good one. However, the deacetylase activity of HDAC11 reported in the literature is very weak or inconclusive, and its ability to deacetylate histones has not been demonstrated. The lack of significant HDAC11 deacetylase activity has truly been a roadblock in this field and has hampered the identification of physiological substrates and development of a highly selective

HDAC11 inhibitor. To the best of our knowledge, previous reported HDAC11 inhibitors are non-specific, although some of them do target mild deacetylase activities. Hening Lin at Cornell University recently developed highly selective HDAC11-specific inhibitors that target HDAC11's removal of long-chain fatty acyl groups from protein lysine residues, and we are currently collaborating with the Lin Lab to test these novel inhibitors in EAE mice. We agree with the editor that these ongoing works are beyond the scope of the current paper.

4. "Why CCL2 reduction was more pronounced in the spinal cord at day 40 than at day 19? ...several mechanisms might have contributed to reduced EAE symptoms."

We agree that potentially more than one mechanism may have contributed to the reduced EAE symptoms in the HDAC11 KO mice, and our discussion reflects this point. What's puzzling to us is the reviewer's question of "Why CCL2 reduction was more pronounced at day 40 compared to day 19." CCL2 is primarily secreted by monocytes and macrophages. At day 40, immune cell infiltration into the spinal cord is clearly reduced compared to day 19 (Figure 2). Our observation that CCL2 reduction was more pronounced in the spinal cord at day 40 than at day 19, is therefore consistent with our data of less demyelination and reduced immune cell infiltration at day 40 compared to day 19, and corresponds to reduced disease severity in the later phase of EAE. We do not think it's necessary to modify any part of our paper to address the reviewer's confusion about this point.

5. "In Fig. 2C, nuclear staining demonstrates severe loss of cellularity even though myelin content (neurofilament, Fig. S4B) is not reduced."

Figure 2 shows spinal cord lesion areas, but Figure S4 shows neurons and neurofilaments in the grey matter. There is no discrepancy. In Figure 2C, the nuclear staining is largely contributed by immune cell infiltration into the spinal cord. This result is consistent with those shown using H & E staining (Figure 2A). Nuclear staining between WT and HDAC11 KO mice are almost identical at day 19. At day 40, however, nuclear staining in HDAC11 KO mice is less than in WT mice, consistent with our data of reduced immune cell infiltration in HDAC11 KO mice.

Reviewer #2

We appreciate Reviewer #2's comment that all previous concerns have been addressed, and that our paper is now suitable for publication.

Reviewer #3

We thank Reviewer #3's comment that "most concerns have been addressed." Our point-by-point response to the reviewer's comments:

1. "...should rephrase the sentence about HDAC11 affects the binding of PU.1 to the CCL2 promoter."

We appreciate the reviewer's suggestion and now replaced sentences related to this, as well as modified the abstract to reflect this important point.

2. "The results do not show that HDAC11 recruits PU.1 to the promoter; instead they show that HDAC11 is required for the binding of PU.1 to that promoter. Different mechanisms, direct and indirect, could explain that requirement..."

We agree with the reviewer's interpretation of our data and changed the discussion to reflect that HDAC11 is required for the binding of PU.1 to the CCL2 promoter. We also appreciate the reviewer's suggestion to elaborate and discuss the possible mechanisms, and we have now expanded the discussion to include this valuable suggestion.

3. "...move the panels in Figure S8 to Figure 7"

Done.

4. "...the new panel S8A shows... only the overexpression condition... Also, the panels S8B and S8C are introduced much earlier than panel S8A."

These issues are now corrected in our manuscript.

September 17, 2018

RE: Life Science Alliance Manuscript #LSA-2018-00039-TRR

Prof. Edward Seto
George Washington University
GW Cancer Center
800 22nd St NW
Room 8800
Washington DC, DC 20052

Dear Dr. Seto,

Thank you for submitting your Research Article entitled "Loss of HDAC11 ameliorates clinical symptoms in a multiple sclerosis mouse model". It is a pleasure to let you know that your manuscript is now accepted for publication in Life Science Alliance. Congratulations on this interesting work.

The final published version of your manuscript will be deposited by us to PubMed Central (PMC) as soon as we are allowed to do so, the application for PMC indexing has been filed. You may be eligible to also deposit your Life Science Alliance article in PMC or PMC Europe yourself, which will then allow others to find out about your work by Pubmed searches right away. Such author-initiated deposition is possible/mandated for work funded by eg NIH, HHMI, ERC, MRC, Cancer Research UK, Telethon, EMBL.

Please also see:

<https://www.ncbi.nlm.nih.gov/pmc/about/authorms/>

<https://europepmc.org/Help#howsubsmanu>

*****IMPORTANT:** If you will be unreachable at any time, please provide us with the email address of an alternate author. Failure to respond to routine queries may lead to unavoidable delays in publication.*******

DISTRIBUTION OF MATERIALS:

Again, congratulations on a very nice paper. I hope you found the review process to be constructive and are pleased with how the manuscript was handled editorially. We look forward to future exciting submissions from your lab.

Sincerely,
